# EVENT-LEVEL CAUSALITY: A FRAMEWORK TO UNIFY CAUSAL MODELING AND EXPLAINABLE AI

## ABSTRACT

Causal relations are typically modeled between random variables (RVs), yet in real-world settings, it is events that cause other events, not RVs causing RVs. We formalize this perspective as Event-Level Causality (ELC), under which Bayesian network structure learning and many Explainable AI (XAI) methods can be viewed as special cases. ELC increases the flexibility of causal modeling by capturing dependencies beyond classical structure learning. It also strengthens XAI by rigorously linking feature importance to causality and showing that different XAI models approximate a principled objective function with varying degrees of fidelity. We propose a new approximation of this objective that, in experiments, clearly outperforms benchmarks such as LIME, L2X, SHAP, and INVASE.

## 1 INTRODUCTION

Causality and explainability are closely related concepts; yet, causal modeling and eXplainable Artificial Intelligence (XAI) have largely developed as separate research fields. A growing body of work seeks to build bridges between these domains (see Carloni et al. (2025) for a recent survey). These pioneering efforts, e.g., using Shapley values in structure learning (Wang et al., 2021) or *do*-calculus (Heskes et al., 2020), serve as conceptual "ropes" connecting the branches of the causal modeling and XAI trees. Nevertheless, the roots of these trees, i.e. their underlying premises, assumptions, and axiomatizations, remain distinct and only partially compatible.

In this work, we aim to fill this gap by providing a flexible unifying framework, which we call *event-level structure learning and causality*, where existing Bayesian network structure learning (BSL) and causal modeling, as well as many XAI approaches, can be viewed as special cases.

Our key observation is that in BSL, dependencies are modeled among random variables (RVs), whereas a subset of XAI approaches, namely amortized explanation methods (AEMs) (Dabkowski & Gal, 2017; Chen et al., 2018; Yoon et al., 2018; Schwab & Karlen, 2019; Jethani et al., 2021), learn a global *selector* model that maps any instance (i.e., input feature values) to a subset of features identified as important for predicting the corresponding output. From a probabilistic perspective, BSL models dependencies at the level of random variables, whereas AEMs capture dependencies at the level of probability events (i.e., realizations of random variables). Another key difference is that an XAI model detects dependencies relevant for predicting a single random variable (namely, the output), whereas BSL models a network of random variables, learning a parent set for each variable subject to the constraint that the resulting directed graph is acyclic.

In our proposed event-level BSL framework, we learn a global selector model for each network random variable, referred to as a *parenthood function*, which returns the parents of that random variable given the realizations of its preceding random variables in the network. Notably, a standard (i.e., RV-level) BSL graph is a special case of the event-level BSL in which the parenthood functions are constant sets.

Under certain assumptions—namely, causal sufficiency, the Causal Markov assumption, and faithfulness—an RV-level BSL graph reflects true causal relationships among random variables (Pearl, 2009; Tonekaboni et al., 2020). By relaxing the Causal Markov assumption, we provide necessary conditions for event-level dependencies to reflect true causes. In real-world settings, it is the events themselves (i.e., what actually happens) that cause other events, rather than random variables causing random variables, and the flow of causes and effects may vary across events. This motivates a

notion of causality that is arguably more natural than the Pearlian definition. Under these relaxed assumptions, event-level BSL defines what we term *event-level causality* (ELC).

ELC is a flexible framework capable of capturing nuanced causal dependencies that go beyond the modeling capacity of existing RV-level causal graphs. It also brings greater rigor to XAI, providing a canonical definition of what "feature importance" means and under what conditions it reflects causality. Such a widely agreed-upon definition of importance is currently missing in the XAI literature, where it is often only implicitly encoded in objective functions.

A canonical definition of feature importance naturally leads to a canonical loss function, which, although abstract, can be instantiated or approximated in various ways, giving rise to different XAI algorithms that share a common high-level objective. As one such instantiation, we propose a concrete XAI model, Hide&Seek, which, in our experiments, outperforms benchmarks including LIME, L2X, SHAP, and INVASE.

## 2 STRUCTURE LEARNING AND CAUSAL DISCOVERY

Let $\mathbf{X}_{1:q} := (X_1, \ldots, X_q)$, be a random vector defined on $\mathcal{X}_{1:q} := \mathcal{X}_1 \times \ldots \times \mathcal{X}_q$. Each $X_i$ may be discrete or continuous. For continuous $X_i$, by $X_i = x_i$ we denote the event $\{x_i - \epsilon < X_i < x_i + \epsilon\}$ for some fixed $0 < \epsilon \ll 1$. This convention prevents measure-theoretic issues such as conditioning on events of zero probability and allows us to treat discrete and continuous random variables similarly. Conditional (in)dependence is a core concept for our discussion. We therefore distinguish between *statistical* and *event-specific*[1] conditional independence:

**Statistical conditional independence.** Random variables $X_i$ and $X_j$ are (conditionally) independent given random variable $X_k$, (denoted $X_i \perp\!\!\!\perp X_j \mid X_k$) if for all $x_i \in \mathcal{X}_i$, $x_j \in \mathcal{X}_j$ and $x_k \in \mathcal{X}_k$

$$P(X_i = x_i, X_j = x_j \mid X_k = x_k) = P(X_i = x_i \mid X_k = x_k) \cdot P(X_j = x_j \mid X_k = x_k). \quad (1)$$

**Event-specific conditional independence.** Random variables $X_i$ and $X_j$ are (conditionally) independent given the event $X_k = x_k$, (denoted $X_i \perp\!\!\!\perp X_j \mid X_k = x_k$) if (1) holds for $X_k = x_k$ and for all $x_i \in \mathcal{X}_i$ and $x_j \in \mathcal{X}_j$.

**Bayesian Structure Learning (BSL).** Statistical conditional independence relations among $\{X_i\}_{i=1}^q$ can be represented using a tuple $\mathcal{G} := \langle pa_1^{(\mathcal{G})}, \ldots pa_q^{(\mathcal{G})} \rangle$, where $pa_i^{(\mathcal{G})}$ represents the set of indices of the parents of $X_i$.[2] The tuple $\mathcal{G}$ is equivalent to a directed graph with nodes $\{X_i\}_{i=1}^q$ and a directed edge from each node in $\mathbf{X}_{pa_i^{(\mathcal{G})}}$ to the node $X_i$. The acyclicity of $\mathcal{G}$ is guaranteed if and only if, with respect to some node ordering, the parents of every node appear earlier in the ordering than the node itself. BSL (as an optimization problem Zheng et al. (2018)) deals with finding the optimal (i.e., the sparsest) factorization of the joint distribution, $P(\mathbf{X})$, through learning a Directed Acyclic Graph (DAG), $\mathcal{G}$, such that: $P(\mathbf{X} = \mathbf{x}) = \prod_{i=1}^q P\left(X_i = x_i \mid \mathbf{X}_{pa_i^{(\mathcal{G})}} = \mathbf{x}_{pa_i^{(\mathcal{G})}}\right)$.

**Causal Structure Learning.** Under the following assumptions, structure learning can recover the true causal relationships among random variables (i.e., $\mathcal{G}$ serves as a *causal graph*):

(I) *Causal Sufficiency*: There are no hidden confounders, and all random variables of interest are measured and included in the dataset.

(II) *Causal Markov Assumption*: The true causal relations are entailed by a unique DAG, implying a set of (statistical) conditional independence relations among the random variables of interest. This DAG structure (and hence the parent set of each node) is fixed and does not change across different realizations of the random variables.

---

[1] *Event-specific conditional independence* is also referred to as *context-specific independence* in the literature.

[2] Note that in the literature, $pa_i^{(\mathcal{G})}$ is usually defined as the set of random variables that are parents of $X_i$; here, we define $pa_i^{(\mathcal{G})}$ as the set of *indices* of those random variables. This allows us to differentiate between $\mathbf{X}_{pa_i^{(\mathcal{G})}}$, the parent random variables of $X_i$, and their realizations, $\mathbf{x}_{pa_i^{(\mathcal{G})}}$.

(III) *Faithfulness Assumption*: The structure of the true causal graph entails all and only the conditional independencies that hold in the joint distribution of the variables of interest. Thus, no conditional dependence is hidden in the distribution due to effect cancellation.

**Known order assumption.** By relying exclusively on observational data, the true causal graph cannot be distinguished from its Markov equivalent graphs. To identify the true causal graph, either there must be a known partial topological ordering among the variables of interest, indicating the direction of causality (e.g, temporal ordering), or sufficient interventional data should be available (e.g., via do-calculus). In this work, we focus on the first scenario. For notational ease, we assume a *full* ordering is known, and the variables are indexed according to that ordering. However, for linking causation to observational data, a known *partial* ordering is sufficient.

Under this assumption, given the parent nodes, $\mathbf{X}_{pa_i^{(\mathcal{G})}}$, of each random variable $X_i$, it is independent of its non-parent *preceding* random variables, $\mathbf{X}_{<i}\backslash\mathbf{X}_{pa_i^{(\mathcal{G})}}$:

$$X_i \perp\!\!\!\perp \mathbf{X}_{<i}\backslash\mathbf{X}_{pa_i^{(\mathcal{G})}} \mid \mathbf{X}_{pa_i^{(\mathcal{G})}}. \tag{2}$$

In the next section, we present event-level structure learning. The known order assumption helps us to interpret directed structural links as causal relations. However, note that in many real-world applications, the Known order assumption is not realistic, and the alternative, that is, collecting interventional data, is not possible either. In such cases, we may learn the ordering by relying on the existing structure-learning algorithms for DAGs, such as NOTEARS Zheng et al. (2018). This resulting approach will still be *event-level structure learning*, in the sense that our algorithm will learn the most parsimonious factorization of a joint distribution per realization of random variables. However, it will lose its causal interpretations, and the direction of edges (other than colliders) will not necessarily show the direction of causes and effects.

## 3 EVENT-LEVEL STRUCTURE LEARNING AND CAUSALITY

In this section, we present our proposed *Event-Level Structure Learning and Causal Modeling* by replacing the *Causal Markov Assumption* (Condition II) of the existing (i.e., RV-Level) BSL with the following less restrictive assumption:

(II⋆) *Event-level Causal Markov assumption*: For each realization $\mathbf{x}$ of the random variables of interest, $\mathbf{X}$ (corresponding to the event $\{\mathbf{X} = \mathbf{x}\}$), the true causal relations are entailed by a DAG which implies a set of *event-specific* conditional independence relations among the random variables. As such, the set of parents of each node $X_i$ is not necessarily fixed and can vary depending on the realization of its predecessors.

To clarify this notion, consider the following running example:

**Example 1** (Modeling Sea & Land Breezes)**.** *Let $T$ be a uniformly distributed random variable representing the time of day, and let the current atmospheric humidity over the Sea ($S$) and Land ($L$) have normal distributions truncated to $[0, 1]$:*

$$T \sim \text{Unif}([0, 24)), \qquad S \sim \mathcal{N}(\mu_1, \sigma_1^2)_{[0,1]}, \qquad L \sim \mathcal{N}(\mu_2, \sigma_2^2)_{[0,1]}.$$

*Random variables $S'$ and $L'$ represent future atmospheric humidity over the Sea and Land, respectively. During the day, the sea breeze transports the sea's atmospheric humidity inland; therefore, both future sea and land humidities, $S'$ and $L'$, depend on the current sea humidity $S$. Conversely, during the nighttime, the current atmospheric humidity over the land, $L$, is advected towards the sea by the land breeze, affecting both $S'$ and $L'$, resulting in the following model:*

$$S' \sim \begin{cases} \mathcal{N}(S, \sigma_3^2)_{[0,1]}, & \text{if } T \in [6, 18] \\ \mathcal{N}(L, \sigma_4^2)_{[0,1]}, & \text{if } T \notin [6, 18] \end{cases}, \qquad L' \sim \begin{cases} \mathcal{N}(S, \sigma_5^2)_{[0,1]}, & \text{if } T \in [6, 18] \\ \mathcal{N}(L, \sigma_6^2)_{[0,1]}, & \text{if } T \notin [6, 18] \end{cases}, \tag{3}$$

*where in the above equations $\mu_1$, $\mu_2$, and $\sigma_1^2$ to $\sigma_6^2$ are fixed hyper-parameters.*

Figures 1a and 1b show that, in the above example, the flow of causes and effects changes for different subsets of the space. Figure 1c illustrates that the static DAG used in classic structure learning models

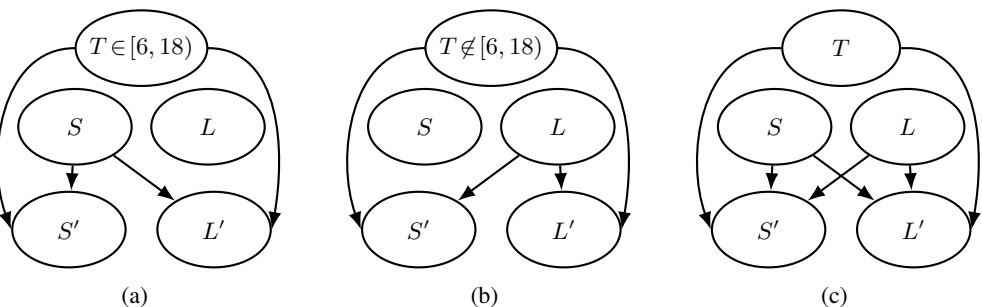

Figure 1: Flow of causes and effects among *Time* ($T \in [0, 24)$), current atmospheric humidity over the *Sea* ($S$) and *Land* ($L$), and future atmospheric humidity over the *Sea* ($S'$) and *Land* ($L'$). (a) Event-level causal graph showing that during the daytime, land humidity is influenced by sea humidity (sea breeze). (b) Event-level causal graph illustrating the nightly land breeze. (c) RV-level Causal graph representing the relations among the random variables.

is not sufficiently flexible to capture such nuanced causal relations. The reason is that event-specific dependence on any subset of the space implies statistical conditional independence. More precisely, if there exists a measurable subset $\sigma \subset \Omega$ with positive probability such that $\forall x_k \in \mathrm{supp}(X_k \mid \sigma)$, $X_i \not\perp\!\!\!\perp X_j \mid X_k = x_k$, then $X_i \not\perp\!\!\!\perp X_j \mid X_k$. Therefore, the causal graph $\mathcal{G}$ that satisfies (2) contains the union of edges from all possible event-specific causal graphs, causing these nuanced causal relations to be lost. This motivates us to formalize a more expressive and flexible alternative to RV-level Bayesian structure learning in Section 3.1.

## 3.1 FORMALIZING EVENT-LEVEL CAUSALITY

We denote a structure that models event-level causality by $\mathcal{H} := (pa_1^{(\mathcal{H})}, \ldots, pa_q^{(\mathcal{H})})$, in which $pa_i^{(\mathcal{H})}$, referred to as the *parenthood function* node $X_i$, is a function that takes a realization $\mathbf{x}_{<i}$ of the nodes that precede $X_i$ and returns the indices of the subset of $\mathbf{X}_{<i}$ that are parents of $X_i$ in that realization.

$$pa_i^{(\mathcal{H})} : \mathcal{X}_1 \times \ldots \times \mathcal{X}_{i-1} \to 2^{\{1, \ldots, i-1\}} \qquad \forall i \in \{2, \ldots q\}.$$

and the joint distribution is factorized as:

$$P(\mathbf{X} = \mathbf{x}) = \prod_{i=1}^{q} P\left(X_i = x_i \mid \{X_k = x_k\}_{k \in pa_i^{(\mathcal{H})}(\mathbf{x}_{<i})}\right). \tag{4}$$

Note that $\{X_k\}_{k \in pa_i^{(\mathcal{H})}(\mathbf{x}_{<i})}$ is not a fixed random variable or vector. Rather, it is a set of random variables (potentially of varying size), where the elements included depend on the realization $\mathbf{x}_{<i}$.

Let $\overline{pa}_i^{(\mathcal{H})}(\mathbf{x}_{<i})$ denote the set of indices of all random variables that precede $X_i$ (for the realization $\mathbf{x}_{<i}$) and are not parents of $X_i$: $\overline{pa}_i^{(\mathcal{H})}(\mathbf{x}_{<i}) := \{1, \ldots, i-1\} \backslash pa_i^{(\mathcal{H})}(\mathbf{x}_{<i})$. Event-level factorization (4) entails that:

$$X_i \perp\!\!\!\perp \mathbf{X}_{\overline{pa}_i^{(\mathcal{H})}(\mathbf{x}_{<i})} \mid \{X_k = x_k\}_{pa_i^{(\mathcal{H})}(\mathbf{x}_{<i})}, \qquad \forall \mathbf{x}_{<i} \in \mathcal{X}_1 \times \ldots \times \mathcal{X}_{i-1}. \tag{5}$$

**Optimal event-level causal structure.** The task of event-level causal modeling is to find a structure $\mathcal{H}^\star$ that satisfies (5) and always returns the minimal set of parents of each random variable; that is, $\mathcal{H}^\star := (pa_1^\star, \ldots pa_q^\star)$ where $\forall \mathbf{x}_{1:q} \in \mathcal{X}_1 \times \ldots \times \mathcal{X}_q$ and $\forall i \in \{2, \ldots, q\}$:

$$pa_i^\star(\mathbf{x}_{<i}) := \underset{pa_i}{\mathrm{argmin}} \|pa_i(\mathbf{x}_{<i})\| \text{ s.t. } X_i \perp\!\!\!\perp \mathbf{X}_{\overline{pa}_i(\mathbf{x}_{<i})} \mid \{X_k = x_k\}_{k \in pa_i(\mathbf{x}_{<i})} \tag{6}$$

Note that the minimal set of parents is not necessarily unique. For example, if $X_3 = \min(1, X_1 + X_2) + \epsilon$ where $\epsilon$ is some independent noise, then for the realization $(X_1 = 1, X_2 = 1)$, given either $X_1$ or $X_2$, $X_3$ becomes independent of the other RVs. Therefore, $pa_3^*(X_1 = 1, X_2 = 1)$, can either be chosen to be $\{1\}$ or $\{2\}$ .

To facilitate optimization, instead of the original objective (6), for each node, $\mathbf{X}_i$, we can minimize the following surrogate loss functions:

$$\ell_i(pa_i, P_{X_i|\mathbb{X}}) = \mathbb{E}_{\mathbf{x}_{<i} \sim \widetilde{P}(\mathbf{X}_{<i})} \Big[ D\big(P^\star(X_i \mid \mathbf{X}_{<i} = \mathbf{x}_{<i}) \parallel P_{\cdot|\mathbb{X}}(X_i \mid \{X_k = x_k\}_{k \in pa_i(\mathbf{x}_{<i})})\big)$$

$$+ \lambda_i \|pa_i(\mathbf{x}_{<i})\| \Big], \qquad \forall i \in \{2, \ldots, q\}, \qquad (7)$$

where (a) $\widetilde{P}(\mathbf{X}_{<i})$ is the empirical distribution of the preceding random variables, $\mathbf{X}_{<i}$; (b) $D(\cdot\|\cdot)$ is a discrepancy measure; (c) $P^\star(X_i \mid \mathbf{X}_{<i})$, the probability of $X_i$ given all preceding random variables, is either learned from the data as a separate optimization task (for measures like KL divergence) or obtained directly from the data (for simpler measures such as absolute error or NLL); (d) $\lambda_i > 0$ is a hyperparameter; (e) the parenthood function $pa(\mathbf{x}_{<i})$ and the conditional probability of $X_i$ given its parents, $P_{\cdot|\mathbb{X}}(X_i \mid \{X_k\}_{k \in pa_i(\mathbf{x}_{<i})})$ (or $P_{X_i|\mathbb{X}}$ for brevity), can be modeled by neural networks with learnable parameters. If (a) removing any subset of the true parents increases the discrepancy by at least $\epsilon$, and (b) removing any RV that is not a true parent, has no effect on the divergence, then since the number of parents of each node $X_i$ is bounded by $(i-1)$, choosing $\lambda_i < \epsilon/(i-1)$ ensures that the losses (6) and (7) have the same minimizer.

Note that from this motivation, we cannot conclude that the hyper-parameter $\lambda_i$ can be arbitrarily small. The reason is that in practice, assumption (b) does not hold, and due to spurious correlations and overfitting, the model (i.e., a Neural Network) can potentially learn to predict the output from non-parent RVs.

In the next section, we link causality with explainability by observing that 7 can be viewed as $(q-1)$ independent instance-wise feature selection XAI problems, and investigating the compatibility of existing XAI methods with this formulation.

## 4 EVENT-LEVEL CAUSALITY AND EXPLAINABLE AI (XAI)

In the event-level causality framework formalized in Section 3.1, let us focus on detecting the parenthood function of a single node, namely node $X_{j+1}$ where $1 < j < q$. For brevity, we introduce new notation: Let $Y := X_{j+1}$ (defined on $\mathcal{Y} := \mathcal{X}_{j+1}$) be an alias for the variable of interest, $\mathbf{X} := (X_1, \ldots, X_j)$ (defined on $\mathcal{X} := \mathcal{X}_1 \times \cdots \times \mathcal{X}_j$) denote the preceding random vector which takes realizations of the form $\mathbf{x} := \mathbf{x}_{\leq j} = (x_1, \ldots, x_j)$. Finally, we denote $pa_{j+1}(\mathbf{x}_{\leq j})$, i.e. the parenthood function of $Y$, by $\mathcal{S}(\mathbf{x}) : \mathcal{X} \to 2^{\{1,\ldots,j\}}$. This setting can be viewed as an XAI problem, where realizations $(\mathbf{X} = \mathbf{x})$ are *feature vectors* and $(Y = y)$ represents the *output*.

Using the new notation, (6) defines the optimal parenthood function of $Y$:

$$\mathcal{S}^\star = \arg\min_{\mathcal{S}} \|\mathcal{S}(\mathbf{x})\| \text{ s.t. } Y \perp\!\!\!\perp \mathbf{X}_{\overline{\mathcal{S}(\mathbf{x})}} \mid \{X_k = x_k\}_{k \in \mathcal{S}(\mathbf{x})}, \qquad \forall \mathbf{x} \in \mathcal{X}. \qquad (8)$$

Roughly speaking, $\mathcal{S}^\star(\mathbf{x})$ returns the indices of the smallest subset of features $\mathbf{x}$ such that, when conditioned on them, $Y$ is independent of the remaining features. By referring to the features $\mathbf{x}_{\mathcal{S}^\star(\mathbf{x})}$ as *(causal) important features*, we obtain a principled definition of XAI "feature importance." We call them *causal* because, under the assumptions of causal sufficiency, faithfulness, known order, and event-level causal Markov, an input feature $x_i$ is a cause of the output if and only if it is important—i.e., $i \in \mathcal{S}^\star(\mathbf{x})$.

### 4.1 COMPATIBILITY OF XAI ALGORITHMS WITH CAUSAL FEATURE IMPORTANCE

Existing instance-wise feature selection XAI algorithms vary in their compatibility with causal feature importance. Local surrogate models like LIME Ribeiro et al. (2016) estimate feature importance via local perturbations. These methods do not necessarily recover features that causally influence the output. This limitation is particularly pronounced when the instance lies far from the decision boundary: For example, if in a model, students with GPA above 3.0 are admitted, perturbing the GPA of a student with GPA 4.0 does not change the output. Consequently, LIME may assign low importance to this student's GPA, even though it is the causal factor. Gradient-based XAI methods, which rely on the model's local derivatives Simonyan et al. (2013); Sundararajan et al. (2017), suffer from the same issue, as gradients vanish for inputs far from the boundary.

Methods that select features by maximizing mutual information with the output (e.g., L2X Chen et al. (2018)) or via Shapley values (e.g., Lundberg & Lee (2017)) define importance in a way that aligns better with our causal feature importance. INVASE Yoon et al. (2018) aligns even more directly, as at a high level, it aims to minimizes a loss equivalent to (7), which in the new notation is:

$$\ell_{\mathbb{X}}(\mathcal{S}, P_{\cdot|\mathbb{X}}) = \mathbb{E}_{\mathbf{x} \sim \widetilde{P}(\mathbf{X})} \Big[ D\big(P^{\star}(Y \mid \mathbf{X} = \mathbf{x}) \parallel P_{\cdot|\mathbb{X}}(Y \mid \{X_k = x_k\}_{k \in \mathcal{S}(\mathbf{x})})\big) + \lambda \|\mathcal{S}(\mathbf{x})\| \Big]. \quad (9)$$

Nonetheless, directly modeling $P_{\cdot|\mathbb{X}}$ is challenging because in each instance, it conditions on a set of potentially different quantities, $\{X_k = x_k\}_{k \in \mathcal{S}(\mathbf{x})}$. INVASE approximates the event $\{X_k = x_k\}_{k \in \mathcal{S}(\mathbf{x})}$ with a fixed-size ablated vector $\{Z_k = x_k \odot \mathbb{I}[k \in \mathcal{S}(\mathbf{x})]\}_{k=1}^{j}$. However, this can lead to an information leak, since the ablated vector contains not only the information of the original event but also additional information from the ablation mask $\{m_k = \mathbb{I}[k \in S(\mathbf{x})]\}_{k=1}^{j}$. In Appendix A, we formally prove that if the output $Y$ depends on an input feature through its partition, the ablation-based approximation of the loss (9) is minimized when the partition with the largest probability mass is ablated. For instance, in Example 1, the only information about $T$ that affects $S'$ and $L'$ is whether it belongs to $[6, 18]$ or not. If more than $50\%$ of the training data points have $T$ in this interval, the selector, $\mathcal{S}$, will learn to ablate $T$ in those cases, and $P_{\cdot|\mathbb{X}}$ will learn to assume $T \in [6, 18]$ whenever its corresponding entry in the ablated vector is 0.

REAL-X (Jethani et al., 2021) avoids INVASE's information leak between the selector and predictor by training them independently. First, the predictor is trained to estimate $Y$ from randomly ablated feature vectors, which prevents information leakage. After the predictor is fixed, the selector is trained to construct instance-wise ablation masks that remove features not affecting the predictor's output. While this approach completely avoids the information leak, it introduces a new challenge: the full model is not end-to-end differentiable, making training complicated and slow. Moreover, the predictor must be sufficiently expressive to predict optimally under arbitrary subsets of features. This is difficult because the number of possible ablation masks grows exponentially with the input dimension $j$. In the next section, we propose an alternative approach that is end-to-end differentiable.

# 5 END-TO-END DIFFERENTIABLE XAI FOR EVENT-LEVEL CAUSALITY

The source of the information leak in the ablation-based methods is that the ablation mask transmits 1 bit of extra information from the selector to the predictor per ablated feature (see Appendix A).

This motivates us to propose a solution: If there is no way for the predictor to differentiate between a signal $\mathbf{x} \sim P(\mathbf{X})$, drawn directly from the original joint distribution and a modified signal, $\mathbf{z}$, coming through the selector, then no information leak is possible. In other words, the modified signal should also be distributed according to $P(\mathbf{X})$, that is, $\mathbf{z} \sim P(\mathbf{X})$.

To construct the modified signal, we keep the values of the RVs selected as important features by the selector, $\mathbf{x}_{\mathcal{S}(\mathbf{x})}$, and replace the remaining RVs with fake values. That is, the modified signal is, $\mathbf{z} := (\mathbf{z}_{\mathcal{S}(\mathbf{x})}, \mathbf{z}_{\bar{\mathcal{S}}(\mathbf{x})}) = (\mathbf{x}_{\mathcal{S}(\mathbf{x})}, \mathbf{x}^{\text{fake}}_{\bar{\mathcal{S}}(\mathbf{x})})$ where, as in Section 4, $\mathcal{S}(\mathbf{x})$ and $\bar{\mathcal{S}}(\mathbf{x})$ denote the indices of the selected and unselected RVs, respectively. To ensure that $P(\mathbf{z}) = P(\mathbf{x})$, we need to draw $\mathbf{x}^{\text{fake}}_{\bar{S}}$ from $P(\mathbf{X}_{\bar{S}} \mid \mathbf{x}_{\mathcal{S}(\mathbf{x})})$. The reason is that, $P(\mathbf{z}) = P(\mathbf{x}_{\mathcal{S}(\mathbf{x})}, \mathbf{x}^{\text{fake}}_{\bar{\mathcal{S}}(\mathbf{x})}) = P(\mathbf{x}_{\mathcal{S}(\mathbf{x})})P(\mathbf{x}^{\text{fake}}_{\bar{\mathcal{S}}(\mathbf{x})} \mid \mathbf{x}_{\mathcal{S}(\mathbf{x})})$ is equal to $P(\mathbf{x}) = P(\mathbf{x}_{\mathcal{S}(\mathbf{x})})P(\mathbf{x}_{\bar{\mathcal{S}}(\mathbf{x})} \mid \mathbf{x}_{\mathcal{S}(\mathbf{x})})$ if and only if $P(\mathbf{x}^{\text{fake}}_{\bar{\mathcal{S}}(\mathbf{x})} \mid \mathbf{x}_{\mathcal{S}(\mathbf{x})}) = P(\mathbf{x}_{\bar{\mathcal{S}}(\mathbf{x})} \mid \mathbf{x}_{\mathcal{S}(\mathbf{x})})$.

While the above condition completely prevents any possible leak between the selector and predictor, in practice, we can relax the condition and draw $\mathbf{x}^{\text{fake}}_{\bar{\mathcal{S}}(\mathbf{x})}$ from $P(\mathbf{X}_{\bar{\mathcal{S}}(\mathbf{x})})$ rather than $P(\mathbf{X}_{\bar{\mathcal{S}}(\mathbf{x})} \mid \mathbf{x}_{\mathcal{S}(\mathbf{x})})$. This can lead to out-of-sample fake draws that, in theory, can transmit information to the predictor. However, unless $\mathbf{X}_{\mathcal{S}(\mathbf{x})}$ and $\mathbf{X}_{\bar{\mathcal{S}}(\mathbf{x})}$ are strongly correlated, it is highly unlikely that the predictor can exploit such dependencies. The latter strategy is equivalent to constructing $\mathbf{z}$ as follows:

$$\mathbf{z} = \mathcal{F}(\mathbf{x}, \mathbf{x}^{\text{fake}}) := \mathbf{x} \odot \mathbf{m} + \mathbf{x}^{\text{fake}} \odot (\mathbf{1} - \mathbf{m}), \text{ where } m_i = \mathbb{I}[x_i \in \mathcal{S}(\mathbf{x})] \text{ and } \mathbf{x}^{\text{fake}} \sim \widetilde{P}(\mathbf{X}).$$

Note that random variables $X_i$ and $X_i^{\text{fake}}$ are independent and identically distributed and therefore have 0 mutual information. This contrasts with perturbation-based approaches such as LIME Ribeiro et al. (2016), which generate strongly correlated random variables that can lead to information leakage,

similar to ablation. We then approximate (9) by,

$$\ell_{\mathbb{Z}}(\mathcal{S}, P_{\cdot|\mathbb{Z}}) := \mathbb{E}_{\substack{\mathbf{x} \sim \widetilde{P}(\mathbf{X}) \\ \mathbf{x}^{\text{fake}} \sim \widetilde{P}(\mathbf{X})}} \Big[ D\big(P^{\star}(Y \mid \mathbf{X} = \mathbf{x}) \parallel P_{\cdot|\mathbb{Z}}(Y \mid \mathbf{Z} = \mathcal{F}(\mathbf{x}, \mathbf{x}^{\text{fake}}))\big) + \lambda \|(\mathcal{S}(\mathbf{x}))\| \Big]. \quad (10)$$

This leads to an XAI approach that is unlikely to suffer from information leakage and approximates the event-level causality loss function (9). We refer to this approach as *Event-Level Causality XAI (ELC-X)*. In Section 5.1, we present a concrete instance of ELC-X.

## 5.1 HIDE&SEEK: A CONCRETE ELC-X ARCHITECTURE

Here we focus on an instance of ELC-X designed for explaining classifiers, where the output $Y \in \{0, 1\}^C$ is a one-hot vector over $C$ classes. We choose cross-entropy as the discrepancy measure $D$, which is natural in this setting. Extension to real-valued outputs is straightforward and can be achieved by replacing the classification loss (cross-entropy) with a regression loss such as Mean Squared Error (MSE).

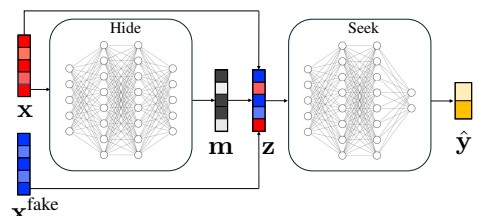

Figure 2: Hide&Seek architecture.

The *Hide&Seek* architecture consists of two feed-forward fully connected neural network modules: *Hide* and *Seek* (see Figure 2).[3] *Hide* takes the input vector $\mathbf{x}$ and produces an *importance mask* $\mathbf{m} \in [0, 1]^j$ corresponding to the selection $\mathcal{S}(\mathbf{x})$. *Seek* maps $\mathbf{z} = \mathcal{F}(\mathbf{x}, \mathbf{x}^{\text{fake}})$ to the predicted output $\hat{\mathbf{y}}$. With these choices, the ELC-X loss function (10) becomes:

$$\ell(\boldsymbol{\theta}, \boldsymbol{\phi}) = -\mathbb{E}_{\substack{(\mathbf{x}, \mathbf{y}) \sim \widetilde{P}(\mathbf{X}, \mathbf{Y}) \\ \mathbf{x}^{\text{fake}} \sim \widetilde{P}(\mathbf{X})}} \left[ \sum_{c=1}^{C} y_c \log \hat{y}_c + \frac{\lambda_s}{j} \|\mathbf{m}\| \right] \quad (11)$$

where $\mathbf{m} = \text{Hide}_{\theta}(\mathbf{x})$, $\hat{\mathbf{y}} = \text{Seek}_{\phi}(\mathbf{z})$, and $\mathbf{z} = \mathbf{m} \odot \mathbf{x} + (1 - \mathbf{m}) \odot \mathbf{x}^{\text{fake}}$ and $\lambda_s$ is a parameter that is progressively increased during training. Specifically, $\lambda_s = \left(\frac{s}{t}\right)^2 \lambda_{\max}$ where $\lambda_{\max} = 0.3$ is a fixed hyperparameter and $s \in \{1, \ldots, t\}$ indexes the training over $t$ epochs.

Experimentally, quadratic growth of $\lambda_s$ significantly improves performance, encouraging the model to prioritize prediction accuracy in the early stages of training and mask sparsity in later stages. Further details on the training procedure and model architecture are provided in Appendix C.

## 6 EXPERIMENTS

We present three analyses. The Synthetic Data experiment evaluates Hide&Seek's ability to recover ground-truth feature importance against existing benchmarks. Switch Analysis evaluates the model's performance in identifying switch-features, that is, features that affect the parenthood functions. The MNIST experiment assesses instance-wise explainability in computer-vision data.

### 6.1 SYNTHETIC DATA

Our first experiment uses the same synthetic datasets as in Yoon et al. (2018), Chen et al. (2018) and Arik & Pfister (2021). The output $Y$ is sampled from a Bernoulli distribution, where $P(Y = 1|U) = \frac{1}{1+e^U}$. $U$ is a function of 11 Gaussian iid variables $X_1, ..., X_{11}$, where $X_i \sim \mathcal{N}(0, 1)$. There are six settings.

The goal is to identify the specific features used in generating $Y$. Syn1-3 represent global feature importance problems. Syn4-6 use a switch feature, $X_{11}$, to create instance-wise feature importance settings. We evaluate each algorithm's success using *True Positive Rate* (TPR) and *False Discovery Rate* (FDR), as in Yoon et al. (2018).

---

[3]Our code is available at `https://github.com/anonymous1861/iclr26_submission`.

- **Syn1**: $U = X_1 X_2$
- **Syn2**: $U = \sum_{i=3}^{6} X_i^2 - 4$
- **Syn3**: $U = -10\sin(0.2X_7) + |X_8| + X_9 + e^{-X_{10}} - 2.4$
- **Syn4**: $U$: if $X_{11} < 0$, follow Syn1, else Syn2
- **Syn5**: $U$: if $X_{11} < 0$, follow Syn1, else Syn3
- **Syn6**: $U$: if $X_{11} < 0$, follow Syn2, else Syn3

We compare seven algorithms. Five are instance-wise feature importance algorithms: Hide&Seek, INVASE, SHAP, LIME and L2X (Yoon et al., 2018; Lundberg & Lee, 2017; Ribeiro et al., 2016; Chen et al., 2018). Two provide global feature importance: LASSO and Random Forest (Tibshirani, 1996; Breiman, 2001). We train on 10,000 samples and test on 10,000 samples.

| | Global Feature Importance | | | | | | Instance-wise Feature Importance | | | | | |
|---|---|---|---|---|---|---|---|---|---|---|---|---|
| Dataset | Syn1 | | Syn2 | | Syn3 | | Syn4 | | Syn5 | | Syn6 | |
| | TPR | FDR | TPR | FDR | TPR | FDR | TPR | FDR | TPR | FDR | TPR | FDR |
| Hide&Seek | **100** | **0** | **100** | **0** | 99 | **0** | **99** | **4** | **97** | 3 | **98** | **4** |
| INVASE | **100** | **0** | **100** | **0** | 96 | **0** | 90 | 10 | 83 | **1** | 90 | 7 |
| SHAP | 62 | 38 | 99 | 1 | 97 | 3 | 67 | 40 | 69 | 40 | 71 | 29 |
| LIME | 20 | 80 | **100** | **0** | 96 | 4 | 54 | 50 | 51 | 54 | 51 | 49 |
| L2X[4] | 18 | 82 | 60 | 40 | 40 | 60 | 46 | 64 | 48 | 61 | 49 | 51 |
| RForest | **100** | **0** | **100** | **0** | **100** | **0** | 67 | 40 | 67 | 40 | 60 | 40 |
| LASSO | 0 | 100 | 50 | 50 | 75 | 25 | 58 | 55 | 56 | 50 | 60 | 40 |

Table 1: Performance of seven algorithms across six datasets. Each metric is the median of 20 experiments. See Appendix D for boxplots.

Hide&Seek outperforms the other models and is fast. INVASE performs well but has a long training time (see Appendix D.2). L2X requires more training data for improved performance (see Appendix D.3). Both INVASE and L2X change the activation function between datasets (ReLU for Syn1-2 and SELU for Syn4-6) (Yoon et al., 2018; Chen et al., 2018). In Hide&Seek, the activation function is always ReLU and the model infrastructure is constant across all datasets. In Hide&Seek and INVASE, any feature with a mask greater than 0.5 is selected. For SHAP, LIME, L2X, LASSO and RForest, the top $k$ important features need to be specified. In the experiments, $k$ is chosen based on the number of ground truth important features sought for each dataset. Specifically, $k = 2$ for Syn1, $k = 4$ for Syn2-3 and $k = 5$ for Syn4-6. This may overestimate the FDR for Syn4 and Syn5, which have only 3 important features when $X_{11} < 0$. To account for this, SHAP, LIME and L2X results for $k = 3$ and $k = 4$ are shown in Appendix D.1.

## 6.2 Switch Analysis

Corollary A.1.1 predicts that the switch-feature misidentification rate of a converged ablation-based XAI algorithm is $50\%$ for the Syn4–6 datasets. To test this, we analyze each model's ability to correctly identify $X_{11}$ as an important feature in the instance-wise synthetic datasets (Syn4-6). As in Section 6.1, we generate 20 independent datasets. We then test how often each model correctly identifies the switch-feature, $X_{11}$. For all datasets, the median switch accuracy of Hide&Seek is $100\%$, significantly higher than that of INVASE and L2X ($\approx 50\%$), and LIME (below $50\%$). The full results in Figure 3 empirically confirm the presence of information leakage in the latter three algorithms.

To assess the benefit of correctly identifying $X_{11}$, we performed a simulation using the results of the INVASE experiments referenced in Table 1. If $X_{11}$ were to be correctly identified, then the TPR would rise to 99.8, 99.7 and 99.9 for Syn4, Syn5 and Syn6, respectively.

## 6.3 MNIST

We use the MNIST dataset to evaluate Hide&Seek in an image recognition setting. The MNIST dataset contains handwritten digits represented as grayscale images (LeCun et al., 2002). As in Chen et al. (2018), we select only the 3s and 8s, with the intention that the minor differences between the two digits might imply feature importance where the ground truth is unknown.

---

[4]L2X has improved performance when trained on more data. Appendix D.3 includes an experiment with training on 1,000,000 samples.

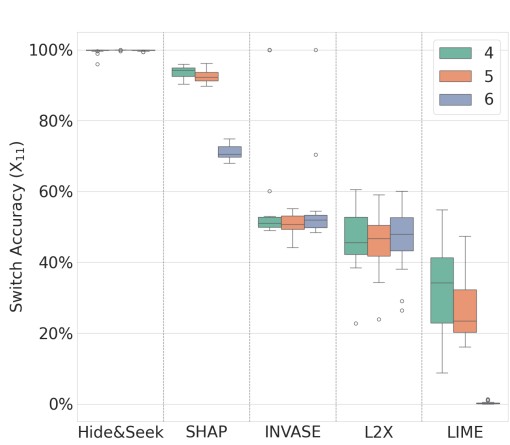

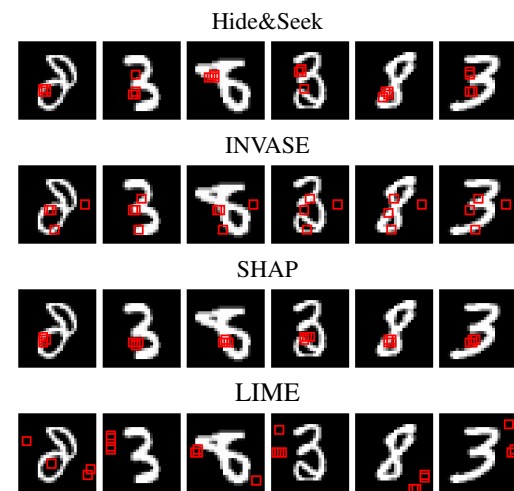

Figure 3: Switch accuracy. The percentage of instances where the switch-feature, $X_{11}$, was correctly identified as important. Each boxplot represents the distribution across 20 runs.

Figure 4: Important patches in MNIST data, as identified by each explanation method. Each 3x3 red square is one of the four most important patches in the image.

We train on 11,172 images and test on 1,397 images. We compare four algorithms: Hide&Seek, INVASE, SHAP and LIME. To correctly assign feature importance, each algorithm first requires a baseline classification rate. For SHAP, we train XGBoost, a tree-based gradient boosting algorithm (Chen & Guestrin, 2016). For LIME we train an independent neural network. Hide&Seek and INVASE perform predictions by design and we use these provided networks without adjusting any architecture. In all four cases, we achieve a classification accuracy of 99.1% or greater. Each image contains $28 \times 28 = 784$ pixels. To identify the most important patches, we assign an importance score to each pixel using the given model. We then apply a $3 \times 3$ sliding window over the image, assigning each $3 \times 3$ patch an aggregated importance score. Figure 4 shows the four most important patches in each image, identified by each model.

Hide&Seek consistently identifies the 'left arcs' of each digit, which indicate an 8 when present and a 3 when absent. Being instance-wise, this model adapts to each drawing and highlights pixels that are important in context. By contrast, INVASE and LIME fail to adapt their patches effectively to the numeral provided.

## 7 RELATED WORK

In the literature, what we term *event-level conditional independence* is usually called *context-specific independence*. We chose the former term to emphasize that in our framework, all random variables can act as *context variables* (a.k.a, selection variables (Pearl & Bareinboim, 2022) or regime indicators (Saggioro et al., 2020)). When these dependencies take simple forms, they can be represented using tree-CPDs (Koller & Friedman, 2009) and Labeled Directed Acyclic Graphs (LDAGs) (Pensar et al., 2015). Algorithms such as (Hyttinen et al., 2018) and (Günther et al., 2024) aim to discover such structures from observational data by proposing extensions to the standard PC algorithm (Spirtes et al., 2000). However, these methods are restricted to discrete context variables. One particular form of context-specific independence that is widely studied includes models with exogenous context variables representing heterogeneous sources of data and subpopulations Forré & Mooij (2018); Zhang et al. (2017); Mooij et al. (2020). Causal discovery in the presence of a single known discrete endogenous context variable has also been proposed Günther et al. (2024). These restrictions are imposed because causal discovery from observational data is challenging, and the true graph can only be detected up to its Markov equivalence class. In this work, we assumed the topological order to be known, which may arise from a temporal ordering, logical constraints, or prior knowledge. This assumption allowed us to remove other restrictions. Specifically, all endogenous variables, whether discrete or continuous, can act as context variables, and no additional constraints on structural

equation models (SEMs) are required. As a result, we provide a unified framework for causal discovery and XAI feature importance methods.

Traditionally, XAI feature importance methods focused on global importance, identifying variables with high predictive contribution across an entire dataset (Tibshirani, 1996; Louppe et al., 2013). More recently, algorithms have been developed for instance-wise (event-level) feature importance, e.g., L2X maximizes the mutual information between the selected features and the output variable (Chen et al., 2018); LIME fits a linear local surrogate model around each input instance (Ribeiro et al., 2016); Integrated Gradients and DeepLIFT are saliency methods that compute the change in the output relative to a baseline for each input feature (Sundararajan et al., 2017; Shrikumar et al., 2017). SHAP uses a game-theoretic approach, averaging the marginal contribution of a feature across all possible feature orderings (Lundberg & Lee, 2017). Other works leverage attention mechanisms for feature importance (Arik & Pfister, 2021; Choi et al., 2016) and develop methods for time-series analysis (Crabbé & Van Der Schaar, 2021; Tonekaboni et al., 2020).

The proposed event-level causality framework should not be confused with *counterfactual reasoning* or *actual causality* Pearl & Mackenzie (2018). See Appendix B for a full discussion. In short, the latter approaches are *backward-looking* Halpern (2016) and based on the abduction of the exogenous values that affected a particular outcome. Our approach, by contrast, is predictive, *forward-looking*, and is not concerned with the exogenous variables that have influenced a specific event.

# 8 CONCLUSION

In this work, we relaxed one of the three assumptions connecting Bayesian Structure Learning (BSL) and causal modeling. Specifically, instead of the standard Causal Markov Assumption—which states that the true causal relations are entailed by a unique set of conditional independence relations among the random variables of interest (represented by a single DAG)—we assumed that for each realization, the true causal relations are entailed by an event-specific DAG. This DAG may vary across events, but under the known order assumption, it respects a topological ordering that constrains the directions of potential causal links.

Under these assumptions, we showed that standard causal structure learning becomes equivalent to a sequence of instance-wise feature selection XAI settings with objective functions of the form (8). We further demonstrated that some existing XAI algorithms, such as INVASE and REAL-X, directly approximate this objective, while others, including L2X and SHAP, optimize related but distinct objectives.

Building on this connection, we proposed a new approximation of (8), termed Event-Level Causality XAI (ELC-X), which provides an abstract formulation capable of incorporating different discrepancy measures. Focusing on classification and using cross-entropy as the discrepancy, we developed a concrete implementation of ELC-X, called Hide&Seek. Our experiments show that Hide&Seek consistently outperforms benchmark algorithms.

We believe that unifying Bayesian structure learning and causal modeling with explainable AI will benefit both fields: the structure learning community can leverage innovations from XAI, while future XAI algorithms can aim to approximate a shared principled objective more effectively.

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

# A   ON FEATURE ABLATION AND INFORMATION LEAK

Consider converting $\{X_k = x_k\}_{k \in \mathcal{S}(\mathbf{x})}$ to a fixed-size vector, $\mathbf{Z}$, via *feature ablation*.

That is, if $k \in \{1, \ldots, j\} \notin \mathcal{S}(\mathbf{x})$, then the corresponding ablated element, $Z_k$, would be a symbol (e.g., $*$) that is not in the original support, $\mathcal{X}_k$.[5] In many XAI methods, $* = 0$ (Zeiler & Fergus, 2014; Petsiuk et al., 2018; Yoon et al., 2018; Schwab & Karlen, 2019); though user-defined values are also common (Ribeiro et al., 2016; Dabkowski & Gal, 2017).

Ablation of features ($\mathbf{X} = \mathbf{x}$) according to the parent index set $\mathcal{S}(\mathbf{x})$ is equivalent to constructing a random vector $\mathbf{Z}$ whose value is the pointwise product of the feature vector $\mathbf{x} := (x_1, \ldots, x_j)$ in a binary *ablation mask* $\mathcal{M} \in \{0, 1\}^j$:

$$\mathbf{Z} = \mathbf{x} \circledast \mathcal{M}(\mathcal{S}(\mathbf{x})) \text{ where } [\mathcal{M}(\mathcal{S}(\mathbf{x}))]_i = \begin{cases} 1, & \text{if } x_i \in \mathcal{S}(\mathbf{x}), \\ 0, & \text{otherwise.} \end{cases} \text{ and } [\mathbf{x} \circledast \mathbf{m}]_i = \begin{cases} x_i, & m_i = 1, \\ *, & m_i = 0. \end{cases}$$

**Theorem A.1.** *Let a subspace, $\mathcal{A}_i \subseteq \mathcal{X}_i$, of a random variable $X_i \in \mathbf{X}$ be partitioned as follows: $\mathcal{A}_i := \bigsqcup_{k=1}^{N} A_{i,k}$ (where $N \geq 1$). Assume $Y$ depends on $X_i$ through its partition i.e., if the realization, $\mathbf{x}_{-i}$, of the other random variables, $\mathbf{X}_{-i} := \mathbf{X} \setminus \{X_i\}$, are given, $Y$ and $X_i$ are dependent but they are independent if the $X_i$'s partition is given:*

$$Y \not\perp\!\!\!\perp X_i \mid \mathbf{X}_{-i} = \mathbf{x}_{-i}, \qquad Y \perp\!\!\!\perp X_i \mid X_i \in A_{i,k}, \mathbf{X}_{-i} = \mathbf{x}_{-i}, \qquad \forall \mathbf{x}_{-i} \in \mathcal{X}_{-i}, \forall A_{i,k} \subset \mathcal{A}_i. \tag{12}$$

*Under this assumption, the loss function of the form:*

$$\ell_{\mathbb{Z}}(\mathcal{S}, P_{\cdot|\mathbb{Z}}) := \mathbb{E}_{\mathbf{x} \sim \widetilde{P}(\mathbf{X})} \Big[ D\big(P^\star(Y \mid \mathbf{X} = \mathbf{x}) \parallel P_{\cdot|\mathbb{Z}}(Y \mid \mathbf{Z} = \mathbf{x} \circledast \mathcal{M}(\mathcal{S}(\mathbf{x})))\big) + \lambda \|(\mathcal{S}(\mathbf{x}))\| \Big]. \tag{13}$$

*is an ill-posed proxy for*

$$\ell_{\mathbb{X}}(\mathcal{S}, P_{\cdot|\mathbb{X}}) = \mathbb{E}_{\mathbf{x} \sim \widetilde{P}(\mathbf{X})} \Big[ D\big(P^\star(Y \mid \mathbf{X} = \mathbf{x}) \parallel P_{\cdot|\mathbb{X}}(Y \mid \{X_k = x_k\}_{k \in \mathcal{S}(\mathbf{x})})\big) + \lambda \|\mathcal{S}(\mathbf{x})\| \Big]. \tag{14}$$

*In the sense that:*

$$\arg \min_{\mathcal{S}} \ell_{\mathbb{Z}}(\mathcal{S}, P_{\cdot|\mathbb{Z}}) \neq \arg \min_{\mathcal{S}} \ell_{\mathbb{X}}(\mathcal{S}, P_{\cdot|\mathbb{X}}). \tag{15}$$

*Proof.* For any ablated vector $\mathbf{x}$, let us define

$$\mathcal{I}(\mathbf{z}) := \{k \in [j] \text{ s.t. } z_k \neq *\}. \tag{16}$$

It is easy to verify that for any parental function $\mathcal{S}$, if $\mathbf{z} = \mathbf{x} \circledast \mathcal{M}(\mathcal{S}(\mathbf{x}))$ and $* \notin \mathcal{X}$, then:

$$\mathcal{I}(\mathbf{z}) = \mathcal{S}(\mathbf{x}), \text{ and } \{z_k\}_{k \in \mathcal{I}(\mathbf{z})} = \{x_k\}_{k \in \mathcal{S}(\mathbf{x})}, \tag{17}$$

because

$$\begin{array}{ll} k \in \mathcal{S}(\mathbf{x}) \implies [\mathcal{M}(\mathbf{x})]_k = 1 \implies z_k = x_k \implies k \in \mathcal{I}(\mathbf{z}) \\ k \notin \mathcal{S}(\mathbf{x}) \implies [\mathcal{M}(\mathbf{x})]_k = 0 \implies z_k = * \implies k \notin \mathcal{I}(\mathbf{z}) \end{array}, \qquad \forall \mathbf{x} \in \mathcal{X}, \forall k \in [j].$$

Let us denote

$$(\mathcal{S}^*, P_{\cdot|\mathbb{X}}^*) = \arg \min_{(\mathcal{S}, P_{\cdot|\mathbb{X}})} \ell_{\mathbb{X}}(\mathcal{S}, P_{\cdot|\mathbb{X}}). \tag{18}$$

We define $P_{\cdot|\mathbb{Z}}^*$ as follows:

$$P_{\cdot|\mathbb{Z}}^*(Y|\mathbf{Z} = \mathbf{z}) := P_{\cdot|\mathbb{X}}^*(Y \mid \{X_k = z_k\}_{k \in \mathcal{I}(\mathbf{z})}) \tag{19}$$

---

[5] If $X_k$ is a continuous random variable and has a Lebesgue density, then the ablated features can be replaced by any prefixed constant $c$, even if $c \in \mathbf{X}_k$, since the probability mass associated with any particular point is 0.

Therefore,

$$P^*_{\cdot|\mathbb{Z}}(Y \mid \mathbf{Z} = \mathbf{x} \circledast \mathcal{M}(\mathcal{S}(\mathbf{x}))) \overset{(19)}{:=} P^*_{\cdot|\mathbb{X}}(Y \mid \{X_k = z_k\}_{k \in \mathcal{I}(\mathbf{z})}) \overset{(17)}{=} P^*_{\cdot|\mathbb{X}}(Y \mid \{X_k = x_k\}_{k \in \mathcal{S}(\mathbf{x})}),$$
(20)

and by substituting (20) in (13):

$$\ell_{\mathbb{Z}}(\mathcal{S}^*, P^*_{\cdot|\mathbb{Z}}) = \ell_{\mathbb{X}}(\mathcal{S}^*, P^*_{\cdot|\mathbb{X}}).$$
(21)

If we show that there is another couple $(\mathcal{S}', P'_{\cdot|\mathbb{Z}})$ where $\ell_{\mathbb{Z}}(\mathcal{S}', P'_{\cdot|\mathbb{Z}}) < \ell_{\mathbb{Z}}(\mathcal{S}^*, P^*_{\cdot|\mathbb{Z}})$, the theorem is proved. Assuming that the premises (12) of the theorem hold, we define

$$P'_{\cdot|\mathbb{Z}}(Y|\mathbf{Z} = \mathbf{z}) := \begin{cases} P^*_{\cdot|\mathbb{X}}(Y \mid \{X_k = z_k\}_{k \in \mathcal{I}(\mathbf{z})}, X_i \in A_{i,1}), & \text{if } z_i = *, \\ P^*_{\cdot|\mathbb{X}}(Y \mid \{X_k = z_k\}_{k \in \mathcal{I}(\mathbf{z})}), & \text{otherwise.} \end{cases}$$
(22)

and

$$\mathcal{S}'(\mathbf{x}) := \begin{cases} \mathcal{S}^*(\mathbf{x})\backslash\{1\}, & \text{if } x_i \in A_{i,1}, \\ \mathcal{S}^*(\mathbf{x}), & \text{otherwise.} \end{cases}$$
(23)

The dependence assumption, $Y \not\perp\!\!\!\perp X_i \mid \mathbf{X}_{-i} = \mathbf{x}_{-i}$, entails that regardless of the realization of $\mathbf{X}$, $X_i$ is a true parent of $Y$. Therefore, $\forall \mathbf{x}, i \in \mathcal{S}^*(\mathbf{x})$:

$$i \in \mathcal{S}^*(\mathbf{x}) \implies [\mathcal{M}(\mathcal{S}^*(\mathbf{x}))]_i = 1 \implies [\mathbf{x} \circledast \mathcal{M}(\mathcal{S}^*(\mathbf{x}))]_i = x_i \neq *, \qquad \forall \mathbf{x}.$$
(24)

If $x_i \notin A_{i,1}$, then,

$$\mathcal{S}'(\mathbf{x}) \overset{(23)}{=} \mathcal{S}^*(\mathbf{x})$$
(25)

Therefore, if $\mathbf{z} = \mathbf{x} \circledast \mathcal{M}(\mathcal{S}'(\mathbf{x}))$, then

$$z_i \overset{(25)}{=} [\mathbf{x} \circledast \mathcal{M}(\mathcal{S}^*(\mathbf{x}))]_i \overset{(24)}{=} x_i \neq * \implies P'_{\cdot|\mathbb{Z}}(Y|\mathbf{Z} = \mathbf{z}) \overset{(22)}{=} P^*_{\cdot|\mathbb{X}}(Y \mid \{X_k = z_k\}_{k \in \mathcal{I}(\mathbf{z})}).$$
(26)

(25) and (26) entail that, as in the previous setting,

$$\ell_{\mathbb{Z}}(\mathcal{S}', P'_{\cdot|\mathbb{Z}}) = \ell_{\mathbb{X}}(\mathcal{S}^*, P^*_{\cdot|\mathbb{X}}), \qquad \forall \mathbf{x} \in \mathcal{X} \text{ s.t.} x_i \notin A_{i,1}.$$
(27)

However, if $\mathbf{x} \in A_{i,1}$ then $\mathcal{S}'(\mathbf{x}) \overset{(23)}{=} \mathcal{S}^*(\mathbf{x})\backslash\{1\}$ which entails:

$$z_i = *, \text{ and } \mathcal{I}(\mathbf{z}) = \mathcal{S}'(\mathbf{x})\backslash\{1\}.$$
(28)

By (28) and (22):

$$P'_{\cdot|\mathbb{Z}}(Y|\mathbf{Z} = \mathbf{z}) = P^*_{\cdot|\mathbb{X}}(Y \mid \{X_k = z_k\}_{k \in \mathcal{S}(\mathbf{x})\backslash\{i\}}, X_i \in A_{i,1}) = P^*_{\cdot|\mathbb{X}}(Y \mid \{X_k = z_k\}_{k \in \mathcal{S}(\mathbf{x})}),$$

in which the last equality holds by the theorem's assumption that $Y$ depends on $X_i$ through its partition.

Therefore, for all $\mathbf{x} \in \mathcal{X}$ such that $x_i \in A_{i,1}$, the divergence terms in (14) and (13) are equal but $\|\mathcal{S}'(\mathbf{x})\| = \|\mathcal{S}^*(\mathbf{x})\backslash\{i\}\| = \|\mathcal{S}^*(\mathbf{x})\| - 1$. This means that on this subset of the space, the instance-wise loss of $(\mathcal{S}', P'_{\cdot|\mathbb{Z}})$ is less than that of $(\mathcal{S}^*, P^*_{\cdot|\mathbb{X}})$, while as we showed by (27), on the rest of space, (where $\mathbf{x} \in \mathcal{X}$ but $x_i \notin A_{i,1}$), $(\mathcal{S}', P'_{\cdot|\mathbb{Z}})$ and $(\mathcal{S}^*, P_{\cdot|\mathbb{X}})$ are associated with the same instance-wise loss. As such,

$$\ell_{\mathbb{Z}}(\mathcal{S}', P_{\cdot|\mathbb{Z}}) < \ell_{\mathbb{Z}}(\mathcal{S}^*, P^*_{\cdot|\mathbb{Z}}) \overset{(21)}{=} \ell_{\mathbb{X}}(\mathcal{S}^*, P^*_{\cdot|\mathbb{X}}).$$

which completes the proof. $\qquad\square$

**Corollary A.1.1.** *If $X_i$ is a random variable with properties* (12)*, then the minimizer of* (13) *ablates $X_i$ whenever $X_i \in A_{i,max} := \arg\max_{A \in \mathcal{A}_i}(\widetilde{P}(A))$, leading to $\frac{P(A_{i,max})}{P(\mathcal{A}_i)}$ misidentification rate.*

*Proof.* In Theorem A.1, any partion in $\mathcal{A}_i$ can be chosen as $A_{i,1}$. Since choosing $A_{i,1} := \arg\max_{A \in \mathcal{A}_i}$ is the minimizer of $\mathbb{E}_{\mathbf{x} \sim \widetilde{P}(\mathbf{X})}\|\mathcal{S}(\mathbf{x})\|$, the conclusion is immediate. $\qquad\square$

## B  EVENT-LEVEL CAUSALITY VERSUS COUNTERFACTUALS AND ACTUAL CAUSALITY (AC)

Pearl & Mackenzie (2018) categorizes the causation into a *Ladder of Causation* with three rungs: Association, Intervention, and Counterfactuals. While the first two rungs study correlations and causality in the level of random variables (type causality), a counterfactual describes a hypothetical scenario where we alter one or more aspects of the world that has "actually" happened (i.e., we alter a given event) and then predict the outcome. More formally, let $\mathbf{X}$ be a random vector that can potentially influence an outcome random variable $Y$. Let the SCM[6] equation that describes $Y$ be $Y = f(\mathbf{X}) + U$, where $U$ is an exogenous variable (noise). Also, let the event that has actually happened be $e := (\mathbf{X} = \mathbf{x}, Y = y)$. In counterfactual reasoning, we are interested in $p(Y_{\mathbf{X} \leftarrow \mathbf{x}} \mid e)$, that is, the probability of the outcome $Y$ in a hypothetical setting were instead of $(\mathbf{X} = \mathbf{x})$, $(\mathbf{X} = \mathbf{x}')$ had happened. For this purpose, firstly, the posterior over exogenous variables given the evidence is computed (general abduction):

$$p(U \mid e) \propto p(U)p(\mathbf{X} = \mathbf{x})p(Y = y \mid \mathbf{X} = \mathbf{x}, U).$$

Once we have $p(U \mid e)$, the counterfactual (under action $do(\mathbf{X} = \mathbf{x}')$) has distribution

$$p(Y_{\mathbf{X} \leftarrow \mathbf{x}'} \mid e) = \int p(Y \mid \mathbf{X} = \mathbf{x}', U)p(U \mid e)dU.$$

A closely related concept is *actual causality* which in the Judea Pearl's framework is defined using counterfactual reasoning. Actual causality tries to determine whether a particular event has been the cause of another occurred event. More concretely, let $M$ be a structural causal model and $u$ a context (realization of random variables). In "Halpern–Pearl definition", in the actual word situation $(M, u)$, an event $\mathbf{X} = \mathbf{x}$ is a cause of an event $Y = y$ iff:

1. AC1 (Actuality): In the actual world situation $(M, u)$, both $\mathbf{X} = \mathbf{x}$ and $Y = y$ occurred:

$$(M, u) \models (\mathbf{X} = \mathbf{x}), \qquad \text{and} \qquad (M, u) \models (Y = y)$$

2. AC2 (Counterfactual dependence under contingencies): Making $\mathbf{X}$ different (while possibly adjusting another set of variables $\mathbf{W}$ to some value $\mathbf{w}$) prevents the effect:

$$(M, u) \models [\mathbf{X} \leftarrow \mathbf{x}', \mathbf{W} \leftarrow \mathbf{w}](Y \neq y)$$

where $[\psi]$ represents an intervention $\psi$ and can alter the model $M$.

3. AC3 (Minimality): $\mathbf{X}$ is minimal – no strict subset of $\mathbf{X}$ satisfies AC1 and AC2.

The main difference of our proposed event-level causality framework with *counterfactual reasoning* and *actual causality* is that both latter approaches are *backward looking* Halpern (2016) i.e., we already know the outcome and we retrospectively ask how it could be different (Counterfactuals) or why it occurred (actual causality) via abduction of the exogenous values.

Our approach, on the other hand, We do not rely on abduction and is not concerned with the exogenous variables of any past event. The proposed event-level causality is *forward looking* and similar to *type causality*, is predictive.

- In actual causality, both causes $(\mathbf{X} = \mathbf{x})$ and effects $(Y = y)$ are events. In contrast, in our approach, the causes are events but the effects are random variables.[7] Consequently, we do not require the AC1 condition, as the effect $Y$ may or may not have occurred (and if it has occurred, i.e., a realization has been assigned to it, this does not affect our reasoning).

- Actual causality AC2 relies on interventions, $[\mathbf{X} \leftarrow \mathbf{x}', \mathbf{W} \leftarrow \mathbf{w}]$. In contrast, interventions are not a necessary element of our approach; instead, we rely on event-specific conditional independence relations.

---

[6]Judea Pearl's Structural Causal Model (SCM) is a Structural Equation Model (SEM) with a causal interpretation.

[7]Compare this to *type causality* where both causes and effects are random variables.

- In actual causality (as well as counterfactual reasoning), the Structural Causal Model (SCM) is assumed to be known – potentially, in the form of a Structural Equation Model (SEM). In contrast, in our approach, this structure is learned from observational data, similar to type causality. Nonetheless, unlike the type causality, our event-level causality defines a model that does not necessarily have a fixed DAG representation (i.e., the topology of the causal graph may change across different events) and can capture more nuanced cause/effect relations.

- In the definition of actual causality, the exogenous variables (a.k.a. context, $u$) are assumed to be fixed. This contradicts Pearl's SEM definition, where the exogenous variables represent all unmodeled influences, and are almost never observable. As such, actual causality is only a semantic definition, not an empirical one. In contrast, our approach is both semantic and practical as it is compatible with the SEM definition and assumes that the exogenous random variables are latent/unknown.

Based on actual causality, Halpern defines another concept called *explanation* (Halpern (2016), Chapter 7). Roughly speaking, $(\mathbf{X} = \mathbf{x})$ is called an explanation of $(Y = y)$ relative to a set of $\mathcal{K}$ of contexts, (in a causal model $M$) if with respect to all contexts in $\mathcal{K}$, $(\mathbf{X} = \mathbf{x})$ serves as an actual cause of $(Y = y)$.

Similar to actual causality, Halpern's *explanation* differs our proposed event-level causality because (1) In Halpern's explanation, the explanandum $(Y = y)$ is an event rather than a random variable; (2) the definition relies on knowing the effect of interventions for all contexts $u \in \mathcal{K}$; and (3) the SEM equations are assumed to be known for all contexts $u \in \mathcal{K}$.

Beckers (2022) is the first work that links actual causality to XAI, or more precisely, to "action-guiding explainable AI", that is, an explainable AI setting where interventions are allowed. In their setting, the relation between $\mathbf{X}$ and $Y$ is deterministic and known in advance, which is a major restriction, but it avoids the problem of conditioning on exogenous random variables (as, effectively, in their setting, there are no exogenous random variables).

To our knowledge, Chockler et al. (2024) is the first and so far only work based on the concept of actual causality that is applicable to an existing XAI problem, namely, image classification.

They assume that all input features are causally independent and, therefore, the interventions reduce to assignments and do not change the black-box structural causal model. This assumption is essential because the overwhelming majority of the existing XAI tasks do not allow for interventions. Nonetheless, this assumption limits the domain of application of this work and makes it immediately irrelevant to the problem of Bayesian Network structure learning, where the random variables are, by definition, allowed to be causally dependent.

Similar to Beckers, Chockler et al. assume that the relation between the input and output is deterministic (whence in effect no exogenous random variables), but unlike the former work, they do not assume that it is known. Instead, to verify (AC2), they need to query the back-box for all possible input values $\mathbf{x}'$, effectively collecting new data. This restricts their approach to discrete input features with small domains that can be effectively enumerated.

Finally, this work cannot predict the black-box output for unobserved input, which is a fundamental limitation of the actual causality framework.

Our proposed XAI framework has none of these limitations. Our approach is a predictive model and learns to predict (as well as explain) via a new deep-learning architecture. It can handle both discrete and continuous features. It applies whether the relationship between input and output is stochastic or deterministic. It learns purely from the available data and requires no interventions or active data collection. Crucially, our method does not assume causal independence among the input features, which is why, when the order of the random variables is known, it can be applied to event-level structure learning.

## C   MODEL INFRASTRUCTURE

**Hide&Seek**. Hide&Seek consists of two fully connected, feed-forward neural networks with ReLU activation functions. The last layer activation function of *Hide* is an element-wise sigmoid to ensure

that the mask $\mathbf{m} \in [0, 1]^j$, while the last layer of *Seek* uses a softmax activation. Each layer has 32 dimensions, and the model is trained for 500 epochs. We use Adam optimizer (learning rate = 0.001), and model weights are initialized using the default PyTorch setting. The implementation is based on PyTorch v2.7.1 with CUDA 12.8.

**INVASE**. The implementation of INVASE uses the code in `https://github.com/iclr2018invase/INVASE`. Specfically, the selector (actor) network has two hidden layers, each with 100 dimensions. The predictor (critic) network has two hidden layers, each with 200 dimensions. The number of training epochs is 10 000, the batch size is 1000 and $\lambda = 0.1$.

**L2X**. The implementation of L2X uses the code in `https://github.com/Jianbo-Lab/L2X/tree/master`. Like INVASE, there are two networks, each with two hidden layers. Each hidden layer of the first network has 100 dimensions and each hidden layer of the second network has 200 dimensions.

**LIME**. The implementation of LIME uses the code in `https://github.com/marcotcr/lime/tree/master`. The baseline models for our Synthetic and MNIST data can be found in our repository.

**SHAP**. The implementation of SHAP uses the code in `https://github.com/shap/shap`. We explored two implementations of the SHAP package: KernelExplainer and TreeExplainer. Kernel SHAP uses weighted linear regression, similarly to LIME (Lundberg & Lee, 2017) and can be run on neural networks. Tree SHAP is a fast, tree-based algorithm that works with ensembles of trees. Tree SHAP performed better on our synthetic data, so we used it with a base XGBoost predictor Chen & Guestrin (2016). The XGBoost model uses the code in `https://pypi.org/project/xgboost/`. The hyperparamters were chosen after some tuning and are: {'objective': 'binary:logistic', 'eval_metric': 'logloss', 'max_depth': 3, 'eta': 0.1, 'num_boost_round': 100}.

**RForest**. The implementation of RForest uses the code in `https://scikit-learn.org/stable/modules/generated/sklearn.ensemble.RandomForestClassifier.html`. The hyperparameters were chosen ofter some tuning and are: {criterion='gini', n_estimators=100, max_depth=5}.

**LASSO**. The implementation of LASSO uses the code in `https://scikit-learn.org/stable/modules/generated/sklearn.linear_model.LogisticRegression.html`. The data are standardized, then run through a logistic regression model with an $L_1$ penalty.

Some minor tweaks to the INVASE and L2X code were made to match our setting and python environments. The updates are contained in our repository.

## D  FURTHER EXPERIMENTS

**Explaining the TPR and FDR metrics.** To compute the TPR and FDR for a given dataset, the TPR and FDR for each data point are calculated using (29). The mean is then taken across the entire dataset. These experiments were conducted 20 times to produce a distribution for the mean TPR, FDR, and F1 are shown in Figure 5. The medians of each boxplot were reported in the paper.

$$\text{TPR} = \frac{\text{true positives}}{\text{true positives} + \text{false negatives}} \qquad \text{FDR} = \frac{\text{false positives}}{\text{true positives} + \text{false positives}} \tag{29}$$

$$\text{F1 score} = 2 \cdot \frac{\text{Precision} \cdot \text{Recall}}{\text{Precision} + \text{Recall}} \tag{30}$$

where Recall = TPR and Precision = $1 - $ FDR.

### D.1  RESULTS FOR $k = 3$ AND $k = 4$

For SHAP, LIME, L2X, LASSO, and RForest, the top $k$ important features need to be specified. In the experiments of the main text, $k$ is chosen based on the number of ground truth important features sought for each dataset. Specifically, $k = 2$ for Syn1, $k = 4$ for Syn2-3 and $k = 5$ for Syn4-6. This

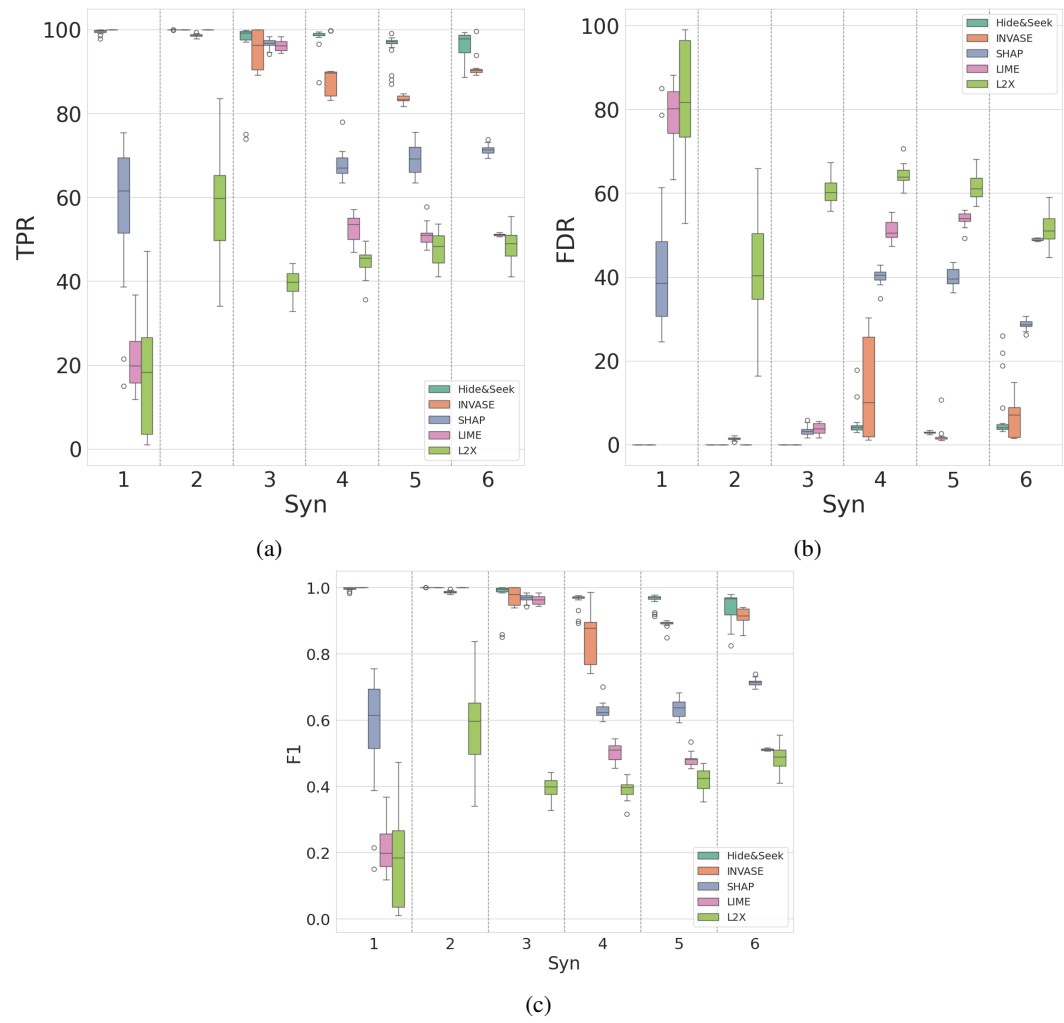

Figure 5: Distributions of (a) mean TPR, (b) mean FDR, and (c) F1 scores for feature identification across Syn1-6. Each boxplot represents 20 experiments. The medians of TPR and FDR boxplots are reported in Table 1.

may overestimate the FDR for Syn4 and Syn5, which have only 3 important features when $X_{11} < 0$. To account for this, SHAP, LIME and L2X results for $k = 3$ and $k = 4$ are shown in Table 2.

| Model | k | Syn4 | | Syn5 | |
|---|---|---|---|---|---|
| | | TPR | FDR | TPR | FDR |
| Hide&Seek | | 99 | 4 | 97 | 3 |
| SHAP | 5 | 67 | 40 | 69 | 40 |
| | 4 | 55 | 39 | 57 | 38 |
| | 3 | 43 | 37 | 44 | 36 |
| LIME | 5 | 54 | 50 | 51 | 54 |
| | 4 | 40 | 50 | 41 | 51 |
| | 3 | 30 | 50 | 30 | 50 |
| L2X | 5 | 46 | 64 | 48 | 61 |
| | 4 | 37 | 64 | 39 | 61 |
| | 3 | 28 | 63 | 31 | 59 |

Table 2: TPR and FDR for Syn4 and Syn5 for different values of $k$, as explained in Section 6.1. Each metric is the median of 20 experiments.

## D.2 RUNNING TIME ANALYSIS

Table 3 reports run times for the instance-wise feature selection models. INVASE's training time is substantially longer than other methods. This might be due to its high model complexity (100k parameters vs Hide&Seek's 3k parameters) and REINFORCE architecture.

|          | Hide&Seek | INVASE   | SHAP        | LIME     | L2X      |
|----------|-----------|----------|-------------|----------|----------|
| Run time | 00:00:05  | 01:20:03 | **00:00:02** | 00:11:29 | 00:00:06 |

Table 3: Average run time (hh:mm:ss) per dataset. Includes training (10,000 samples), predicting and assigning feature importance (10,000 samples). All algorithms are run on the same hardware described in Appendix D.7.

## D.3 TRAINING ON 1,000,000 SAMPLES

Table 4 shows the results from a single experiment for each model and dataset, where the training data was 1,000,000 samples. With a larger training size, L2X has improved results.

| Dataset | Global Feature Importance | | | | | | Instance-wise Feature Importance | | | | | |
|---------|------|------|------|------|------|------|------|------|------|------|------|------|
|         | **Syn1** | | **Syn2** | | **Syn3** | | **Syn4** | | **Syn5** | | **Syn6** | |
|         | TPR | FDR | TPR | FDR | TPR | FDR | TPR | FDR | TPR | FDR | TPR | FDR |
| Hide&Seek | **100** | **0** | **100** | **0** | 99 | **0** | **100** | **1** | **98** | **1** | **98** | **1** |
| INVASE    | **100** | **0** | **100** | **0** | 90 | **0** | 89 | **1** | 84 | **1** | 90 | **1** |
| SHAP      | 98 | 2 | **100** | **0** | **100** | **0** | 67 | 39 | 67 | 39 | 75 | 25 |
| LIME      | 24 | 76 | **100** | **0** | 98 | 2 | 55 | 49 | 50 | 53 | 51 | 49 |
| L2X       | **100** | **0** | **100** | **0** | 86 | 14 | 90 | 27 | 92 | 28 | 88 | 12 |
| RForest   | **100** | **0** | **100** | **0** | **100** | **0** | 67 | 39 | 67 | 39 | 60 | 40 |
| LASSO     | 0 | 100 | 25 | 75 | 75 | 25 | 37 | 70 | 73 | 40 | 50 | 50 |

Table 4: Performance of seven algorithms across six datasets when trained on 1,000,000 samples.

## D.4 TRAINING ON 100 FEATURES

We conduct an experiment in which we increase the number of synthetic features from 11 to 100. The relationship between features remains as described in Section 6.1. There are now an additional 89 noise signals. Hide&Seek outperforms SHAP and L2X but not INVASE.

| Dataset | Global Feature Importance | | | Instance-wise Feature Importance | | |
|---------|------|------|------|------|------|------|
|         | Syn1 | Syn2 | Syn3 | Syn4 | Syn5 | Syn6 |
| Hide&Seek | 86 | 97 | 98 | 65 | **75** | 72 |
| INVASE    | **100** | **100** | **100** | **71** | 64 | **78** |
| SHAP      | 8 | 98 | 96 | 58 | 56 | 71 |
| L2X       | 0 | 6 | 4 | 5 | 5 | 5 |

Table 5: F1 values for high-dimensional (100 features) synthetic datasets. The target is still the same function of $\{X_1, \ldots, X_{11}\}$ as in the earlier experiments. But in this experiment includes 89 extra unimportant features (independent noise). Each F1 value is the median of 20 experiments.

## D.5 SYN3 - SPECIFICATION VS. IMPLEMENTATION

We note a minor discrepancy between the specification of model Syn3 in the previous works (Yoon et al., 2018; Chen et al., 2018; Arik & Pfister, 2021), and the code linked in their publication.

| Method | Source | Syn3 |
|--------|--------|------|
| INVASE | Paper | $-10\sin(2X_7) + 2|X_8| + X_9 + \exp(-X_{10})$ |
|        | Code  | $-10\sin(0.2X_7) + |X_8| + X_9 + \exp(-X_{10}) - 2.4$ |
| L2X    | Paper | $-100\sin(2X_1) + 2|X_2| + X_3 + \exp(-X_4)$ |
|        | Code  | $-100\sin(0.2X_1) + |X_2| + X_3 + \exp(-X_4) - 2.4$ |
| Hide&Seek | Both | $-10\sin(0.2X_7) + |X_8| + X_9 + \exp(-X_{10}) - 2.4$ |

Table 6: Paper and code expressions for Syn3 model in INVASE, L2X, and Hide&Seek.

For our experiments, we use the data-generating model of the previous code, so that the results are comparable.

### D.6 CALIFORNIA HOUSING DATA

In this experiment, we evaluate the models against correlated real-world features. The California Housing dataset contains information on housing block groups in California from the US 1990 Census (Pace & Barry, 1997). Each block group represents an average of 1425.5 people living in proximity. There are 20,640 block groups and 8 features: *Median Income, Median House Age, Average Rooms, Average Bedrooms, Population, Occupancy* (average members per household)*, Latitude* and *Longitude*.

We generate the target synthetically to establish a ground-truth feature importance against which to evaluate our models. As in the synthetic data experiments, we simulate a classification problem where $Y$ is sampled from a Bernoulli distribution and $P(Y = 1|U) = \frac{1}{1+e^U}$. We use a switch feature for three geographic locations.[8] In the first geography, $U = $ Average Rooms $-$ Average Bedrooms; in the second geography, $U = \frac{\text{Population}}{\text{Occupancy}}$; and in the third geography $U = 5 \times$ Median Income $- 2 \times$ (Median House Age)$^2$.

The results are shown in Tables 7 and 8. Hide&Seek and INVASE have a natural mechanism to determine if a feature is important, namely $m_i > 0.5$. However, the other models require specifying the number of desired features, $k = 3$. In this experiment there are three important features for all houses.

| Model | Accuracy (%) |
|-------|--------------|
| Hide&Seek | **76** |
| INVASE | 29 |

Table 7: Percent of 2,064 test samples where all three true features were identified and none were incorrectly identified. Both Hide&Seek and INVASE have mechanisms for determining feature importance without needing to specify how many features are required.

| Model | Accuracy (%) |
|-------|--------------|
| Hide&Seek | **85** |
| INVASE | 38 |
| L2X | 10 |
| SHAP | 5 |
| LIME | 1 |

Table 8: Percent of 2,064 test samples where all three true features were correctly identified when the top three features were selected.

### D.7 HARDWARE

Experiments were conducted on the following hardware:

- AMD EPYC 9354P 3.25GHz 32 cores 256MB L3 Cache (Max Turbo Freq. 3.75GHz)
- 192GB 4800MHz ECC DDR5-RAM (Twelve Channel)
- 1.92TB NVMe SSD Drive and 1.92TB NVMe SSD Drive
- 2x NVIDIA L4 (7,680 Cores, 240 Tensor Cores, 24GB Memory) GPUs

---

[8]Geography 1: Longitude $< -121$. Geography 2: $-121 \leq$ Longitude $< -118$. Geography 3: Longitude $\geq -118$.

