# OpenReview forum: "Event-Level Causality: A Framework to Unify Causal modeling and Explainable AI"
_ICLR.cc/2026/Conference — Submitted to ICLR 2026_

### Official Review · Reviewer_azxR · 2025-10-26

**Soundness:** 2
**Presentation:** 3
**Contribution:** 2
**Rating:** 4
**Confidence:** 3

**Summary:**

This paper introduces Event-Level Causality (ELC), a framework proposing that causal relationships occur between specific events (realizations of random variables) rather than solely between the random variables themselves. ELC generalizes traditional Bayesian Structure Learning by allowing parent sets ('parenthood functions') to vary depending on the specific realization of preceding variables. The authors argue ELC provides a more natural view of causality and bridges causal modeling with XAI, particularly instance-wise feature selection methods. Under ELC, many XAI methods approximate a principled objective for identifying the minimal set of features rendering an output conditionally independent of others given an instance. A new XAI method, Hide&Seek, is proposed based on this objective, designed to avoid information leakage issues present in some prior methods (e.g., INVASE).

**Strengths:**

1. The core idea of formalizing causality at the event level is thought-provoking and offers a potentially richer, more flexible alternative to standard RV-level causal graphs, especially for capturing context-specific dependencies.

2. Hide&Seek demonstrates superior performance compared to established XAI benchmarks on synthetic tasks designed to test global, instance-wise, and context-specific feature importance recovery.

**Weaknesses:**

1. The paper relaxes the Causal Markov Assumption but retains Causal Sufficiency, Faithfulness, and Known Order. The practical implications and limitations of relying on these assumptions in the context of event-level variability need further discussion, especially regarding identifiability guarantees beyond simple cases.

2. The scalability of learning a full ELC structure (beyond the single-node XAI setting) is not explored.

3. The relationship between ELC and existing work on context-specific independence (CSI), labeled DAGs, and tree-CPDs could be elaborated further.

**Questions:**

1. Under what precise conditions can the optimal event-level parenthood functions $\mathcal{S}^{*}(x)$ be uniquely identified from observational data, given the known order assumption but allowing for event-level graph changes? How does this compare to identifiability in standard BSL?

2. How well does the proposed Hide&Seek loss (Eq. 11), using fake draws, approximate the theoretical ELC objective (Eq. 9)? Could the discrepancy impact the identification of the truly minimal parent sets in complex dependency scenarios?

**Details Of Ethics Concerns:**

I did not find Section "THE USE OF LARGE LANGUAGE MODELS" in this paper. This would violate ICLR submission rules and should be desk-rejected.

---

> ### Author Response · Authors · 2025-12-04
> **In response to Reviewer azxR**
>
> We thank the reviewer for their comments are review.
>
> >Q. Under what precise conditions can the optimal event-level parenthood functions be uniquely identified from observational data, given the known order assumption but allowing for event-level graph changes?
>
> A.
> The event-level parenthood functions are uniquely identifiable from observational data under the known order assumption when each *true* parent contributes unique predictive information about the child that cannot be fully replicated by any subset of the other parents.
>
> As an instance of the case where this condition fails, consider $Y = \min(1, X_1+X_2) + \epsilon$ with $\epsilon$ being independent noise.
> In this case, for any event $e$ where $X_1$ and $X_2$ are both greater than or equal to 1, either $\\{X_1\\}$ or $\\{X_2\\}$ can be treated as minimal parents of $Y$ for the event $e$.
>
> We have clarified this in lines 213-215 of the updated manuscript.
>
> >Q. (continued) How does this compare to identifiability in standard BSL?
>
> A. The theoretical link between RV-level and event-level parenthood is as follows: an RV $X$ is an RV-level parent of $Y$ if there exists a positive-probability event, $e$, for which $X$ is an event-level parent of $Y$.
>
> Event-level identifiably is sufficient (but not necessary) for RV-level identifiably. For instance, in the above example, if events $e_1=(X_1=0, X_2=1)$ and $e_2=(X_1=1, X_2=0)$ have positive probability, then the RV-level parents are uniquely identifiable as $\\{X_1, X_2\\}$.
> However, if $X_1$ and $X_2$ are always simultaneously greater than 1 or less than 1, then even in the RV-level, the parents (or more precisely, the minimal set of sufficient causes) is not unique.
>
> Finally, note that this discussion is not related to the lack of identifiably due to Markov Equivalent Classes (MECs), which is ruled out by the known order assumption.
>
> >Q. How well does the proposed Hide and Seek loss (Eq. 11), using fake draws, approximate the theoretical ELC objective (Eq. 9)?
>
> A. It performs quite well, except in situations where the relevant  information carried by some features is highly correlated (e.g. two feature are almost deterministically related).
> We discuss this issue in lines 300-320 of the updated manuscript.
>
>
> > (continued) Could the discrepancy impact the identification of the truly minimal parent sets in complex dependency scenarios?
>
> A. Indeed, if two RVs contain almost identical predictive information, then identifying which one is the *true* cause based solely on observational data is a major challenge for any XAI approach.
>
>
> > I did not find Section "THE USE OF LARGE LANGUAGE MODELS" in this paper. This would violate ICLR submission rules and should be desk-rejected
>
> A. In the OpenReview submission interface, once we selected the option “We have not used LLMs’’, the system automatically disabled the checkbox corresponding to the requirement to include the section “THE USE OF LARGE LANGUAGE MODELS’’. Therefore, the absence of that section in our submission is fully compliant with the submission rules.

---

### Official Review · Reviewer_r2c6 · 2025-10-26

**Soundness:** 3
**Presentation:** 3
**Contribution:** 3
**Rating:** 6
**Confidence:** 3

**Summary:**

This paper proposes a framework for event-level structure learning and causal modeling. By relaxing the traditional causal Markov assumption, it allows each event to correspond to a potentially different DAG structure. This framework unifies traditional random variable-level causal discovery with instance-level feature importance interpretation methods. The authors further propose a specific XAI model called Hide&Seek, which effectively avoids information leakage through a dual-network architecture and feature replacement mechanism. Experiments show that Hide&Seek outperforms existing baseline methods in both feature selection accuracy and the ability to identify "switched features" on synthetic data and the MNIST image recognition task.

**Strengths:**

The ELC framework, first proposed, breaks down the barriers between causal modeling and XAI from an "event-level" perspective. It unifies BSL and most XAI methods as special cases, resolving the core issue of incompatible underlying assumptions between the two fields and providing a foundational theoretical framework for subsequent cross-disciplinary research.

By replacing the traditional fixed DAG assumption with the "event-level causal Markov assumption," the set of parent nodes can dynamically adjust with events. This successfully captures dynamic causal dependencies that traditional RV-level modeling cannot, better aligning with the reality of "causal relationships changing with the scenario." The method is ingeniously designed. The Hide & Seek model effectively avoids information leakage through feature replacement, boasts a simple structure, and is end-to-end differentiable.

The method is well-validated experimentally, with a systematic evaluation conducted on various synthetic data sets and MNIST. The method surpasses seven baseline models in key metrics and training efficiency, yielding compelling empirical results.

The theoretical analysis of XAI methods is in-depth, and the appendix formally analyzes the information leakage issues of existing XAI methods. This paper provides a canonical definition of XAI "feature importance" based on causal relationships, which makes up for the defects of existing XAI methods in which "feature importance" is mostly implicitly defined and lacks causal anchoring, and upgrades XAI explanations from "statistical correlation" to "causally explainable".

**Weaknesses:**

The paper explicitly relies on the "known topological order of variables" to distinguish causal directions, but fails to discuss solutions when this assumption breaks. This limits the applicability of the ELC framework to complex real-world problems.

Generalization to complex data scenarios is not fully verified: The MNIST experiments focus only on two simple classification tasks, "3 vs. 8," and fail to verify the model's performance on multi-class (such as 10-class MNIST), high-dimensional data (such as natural language processing and medical imaging), or data with noise or outliers, raising questions about its generalization ability.

The paper fails to discuss the applicability of Hide&Seek to non-classification tasks, indicating insufficient task coverage. The paper uses the ε interval approximation for continuous variables, which avoids measure-theoretic issues but may introduce approximation errors.

A lack of comparative analysis with "counterfactual XAI": The paper briefly distinguishes the "forward/backward" differences between ELC and counterfactual reasoning, but fails to compare their advantages and disadvantages in practical application metrics such as "explanation credibility" and "user acceptance," nor does it explore the possibility of combining ELC with counterfactual methods. Although Appendix B attempts to distinguish between event-level causality and counterfactual reasoning, the boundaries and complementarity between the two in practical applications could be further clarified.

**Questions:**

How can we extend the event-level causal framework when no topological order is known? Can we consider partial order or use intervening data?

Does Hide&Seek work with high-dimensional or structured data (e.g., images or text)? Does it need to be combined with convolutional or attention mechanisms? How interpretable are event-level causal graphs in real-world systems? Might they be difficult to understand due to their complex structure?

Can event-level causality be combined with counterfactual reasoning? For example, given a known SCM, can event-level parent node sets be used for more accurate counterfactual inference?

---

> ### Author Response · Authors · 2025-12-03
> **In response to Reviewer r2c6 (part 1)**
>
> We thank the reviewer for their insightful comments and their positive evaluation of our work.
>
> > On assessing the performance on multi-class classification scenarios, high-dimensional data, etc.
>
> A.
> The datasets that we have tested against come from the existing XAI literature e.g., the use of two classes: “3 vs. 8” in MNIST data is in line with the MNIST experiment of the seminal paper ‘Learning to Explain' (L2X) (Jianbo Che et al 2018). This choice was made to enable an intuitive evaluation of explainability, where the ground truth is unknown, by comparing the similar-looking numbers 3 and 8.
>
> To address the reviewer's comment, we have also tested the performance of our approach on two new models.
>
> Firstly, we have tested Hide-and-Seek on a multi-class classification task using the California Housing dataset, which includes highly correlated features taken from an existing real-world open database.
> On this dataset, Hide and Seek outperforms other models in identifying feature importance. Specifically, the percentages of houses where perfectly correct feature importance is achieved are:
>
> Hide and Seek: $85 \\%$
>
> INVASE: $38\\%$
>
> L2X: $10\\%$
>
> SHAP: $5\\%$
>
> LIME: $1\\%$
>
> Please see Appendix D.6 in the updated manuscript.
>
> (2) Second, in line with the reviewer’s suggestion, we have extended the synthetic datasets to 100 random variables and found that Hide and Seek outperforms INVASE, REAL-X, L2X, and SHAP. Please see Table 5 in Appendix D.4 of the updated manuscript.
>
>
> >The paper fails to discuss the applicability of Hide and Seek to non-classification tasks
>
> A. Thanks for raising this point. In lines 339-342 of the updated manuscript, we now clarify that "Extension to real-valued outputs is straightforward and can be achieved by replacing the classification
> loss (cross-entropy) with a regression loss such as Mean Squared Error (MSE)."
>
> Our choice to focus on classification in this paper is in line with parallel papers in the field. Namely, INVASE, L2X, and Realx all focused on classification in their papers.
>
>
>
> >Q. How can we extend the event-level causal framework when no topological order is known? Can we consider partial order or use intervening data?
>
> A. Thank you for raising this important point. We have now addressed it in lines 112–132 of the updated manuscript.
> In short:
>
> 1. What is required is a known partial topological order that specifies the direction of potential causes and effects. A cause of a node must precede that node in the (partial) order, but not every node that precedes a given node is necessarily its cause. Nodes that are incomparable in the partial order cannot be causes or effects of one another. Our assumption of a full known ordering was made purely for notational convenience.
>
> 2. If such a partial order is not known but interventions are available, then the ordering can be recovered directly through interventional data, as the reviewer suggests.
>
> 3. If interventions are not possible, the ordering may be inferred using existing DAG structure-learning algorithms, such as NOTEARS (Zheng et al., 2018). In this case, however, the resulting method should be viewed as event-level structure learning, meaning that the algorithm learns the most parsimonious factorization of the joint distribution for each realization.
> The causal interpretation is lost, and the direction of edges (except for colliders) will not reliably correspond to true causal directions.
>
>
> >Q. Does Hide and Seek work with high-dimensional or structured data (e.g., images or text)? Does it need to be combined with convolutional or attention mechanisms?
>
> A. Hide and Seek has outperformed the benchmark algorithms on every model we have tested. To retain comparability with similar algorithms (INVASE, Realx), we did not include convolutional layers or attention mechanisms. Our MNIST experiment was on $28 \\times 28$ images, resulting in 784-dimensional data. In this experiment, all predictive models had a prediction accuracy larger than $99\\%$, which we deemed a strong enough base upon which to analyse explainability. It is worth noting that the ‘Hide’ module in our framework not only enabled explainability but also increased the prediction accuracy in our MNIST experiment. The ‘Seek’ neural network, on its own, achieves $98.9\\%$ accuracy when predicting. But, when coupled with the ‘Hide’ module that masks unnecessary data, the prediction accuracy rises to $99.1\\%$.
>
> >Q. How interpretable are event-level causal graphs in real-world systems? Might they be difficult to understand due to their complex structure?
>
> A. For any given event, the event-level causal graph is simply a DAG and is no more complex than the graphs produced by existing structure-learning methods. Its interpretability is therefore on par with standard DAG-based representations.

---

> ### Author Response · Authors · 2025-12-03
> **In response to Reviewer r2c6 (part 2)**
>
> >Q. Can event-level causality be combined with counterfactual reasoning? For example, given a known SCM, can event-level parent node sets be used for more accurate counterfactual inference?
>
> A. If the underlying SCM is already known, then predictive causal structure learning—whether at the RV level or the event level—becomes unnecessary.
> A more interesting and practically relevant setting is when the SCM is not known and must be learned from data. In that case, one can first learn the structure (either at the RV level or the event level) and then perform counterfactual reasoning on the learned model.
>
> In this latter setting, we believe event-level predictive causality can be beneficial. Counterfactual reasoning, particularly from a Bayesian perspective, involves:
> (1) Computing the posterior distribution over exogenous variables, and
> (2) Evaluating how the output would change under alternative inputs by marginalising over these exogenous variables using their posterior.
>
> Because event-level causality provides a more fine-grained and nuanced characterisation of dependencies, we expect that the resulting posterior over the exogenous variables will often be more concentrated (i.e., closer to deterministic). This, in turn, should improve the accuracy of counterfactual predictions. We consider this a promising direction for future research.
>
> That said, many works that use the term counterfactuals adopt the framework of Halpern’s actual causality, which is more logic-oriented than probabilistic. Even when probabilistic relaxations are used, these definitions typically do not rely on explicit exogenous random variables as the source of uncertainty (e.g., they use statements of the form $Pr((M,u) \models \varphi) > \text{cnst.}$). Such probabilities are treated as primitives rather than quantities learned and updated from data via Bayes’ rule.
>
> In Appendix B of the updated manuscript, we provide a more detailed comparison between our approach and Halpern’s actual causality framework.

---

### Official Review · Reviewer_rgxT · 2025-10-31

**Soundness:** 2
**Presentation:** 2
**Contribution:** 2
**Rating:** 6
**Confidence:** 3

**Summary:**

This paper proposes an ambitious and conceptually rich framework called Event-Level Causality (ELC), which seeks to bridge the gap between causal structure learning and explainable AI (XAI). The central argument is that traditional Bayesian structure learning (BSL) models dependencies between random variables, while real-world causation operates between events. The authors formalize this idea into an event-level causal model and propose a practical instantiation, Hide&Seek, a differentiable XAI model that approximates causal feature importance. The paper is theoretically sound, well-motivated, and relevant to both the causal inference and explainability communities. It demonstrates a deep understanding of the limitations of current causal and interpretability paradigms and offers a unifying framework that could inspire a new direction of research.

**Strengths:**

1. The formal relaxation of the Causal Markov Assumption to the event-level causal Markov assumption is novel and well-argued.
2. The framework elegantly unites Bayesian structure learning and instance-wise XAI under a single formalism.
3. Definitions (e.g., event-specific conditional independence) and derivations are mathematically precise and insightful.
4. The proposed Hide&Seek model is well-designed, end-to-end differentiable, and empirically validated.

**Weaknesses:**

1. The model assumes a known topological order among variables. This assumption is strong and unrealistic for most real-world problems where causal direction is unknown. The framework would benefit from a discussion of how to relax or learn such orderings.
2. Experiments rely primarily on synthetic data and MNIST, both low-dimensional and noise-free contexts. The model’s scalability to high-dimensional or tabular real-world datasets (e.g., genomics or finance) is untested.
3. While theoretically elegant, event-level causality may be difficult to operationalize or interpret in practice. The causal semantics of varying DAGs per event may not align with standard structural causal models (SCMs).
4. The paper positions ELC as a bridge to causal modeling, but it does not evaluate causal metrics (e.g., recovering causal parents or do-interventions). This leaves uncertainty as to whether the learned relationships are truly causal or just predictive.

**Questions:**

1. How can the known ordering assumption be relaxed? Could the framework integrate structure-learning techniques (e.g., NOTEARS) to infer orderings?
2. Does event-level causality preserve causal identifiability under interventions?
3. Could the event-level parenthood function be interpreted as a probabilistic distribution over potential causal structures?
4. How robust is Hide&Seek to correlated features or spurious dependencies?

---

> ### Author Response · Authors · 2025-12-03
> **In response to Reviewer rgxT**
>
> We thank the reviewer for their insightful comments and their positive evaluation of our work.
>
> >Q. How can the known ordering assumption be relaxed? Could the framework integrate structure-learning techniques (e.g., NOTEARS) to infer orderings?
>
> A. Our proposed approach can be combined with NOTEARS or similar methods to learn the variable ordering. However, such a combination should be considered "event-level structure learning" (i.e., event-level minimal factorization of a joint distribution) rather than "event-level causality," since algorithms like NOTEARS rely only on observational data and therefore can establish causality only up to a Markov equivalence class. As a result, the direction of arrows (except for colliders) does not necessarily indicate the true causal direction.
> Another approach to learn the order of random variables is via interventions, if the model being explained permits interventions. We have discussed these possibilities in lines 112-132 of the updated manuscript.
>
> >Q. Experiments rely primarily on synthetic data and MNIST, both low-dimensional and noise-free contexts.
>
> A. First, note that our synthetic models do include noise, as the output $Y$depends on the input $X$ via a Bernoulli distribution. Second, in line with the reviewer’s suggestion, we have extended the synthetic datasets to 100 random variables and found that Hide and Seek outperforms INVASE, REAL-X, L2X, and SHAP. Please see Table 5 in Appendix D.4 of the updated manuscript.
>
> We have also tested Hide-and-Seek on the California Housing dataset, which includes highly correlated features. The features are taken from an existing real-world open database.  To ensure ground-truth feature importance is known, we create a synthetic target. In this semi-synthetic setting, Hide and Seek outperforms other models in identifying feature importance. Specifically, the percentages of houses where perfectly correct feature importance is achieved are:
>
> Hide and Seek: $85 \%$
> INVASE: $38\%$
> L2X: $10\%$
> SHAP: $5\%$
> LIME: $1\%$
>
> Please see Appendix D.6 in the updated manuscript.
>
> >Q. Does event-level causality preserve causal identifiably under interventions?
>
> A. Yes. Even without interventions, causal identifiability is preserved under our assumptions, including the known order assumption, causal sufficiency, causal Markov assumption, and faithfulness assumption. Furthermore, if interventions are allowed, not only event-level causality still preserves causal identifiability, but the known order assumption can be relaxed, as the node ordering can be learned directly via interventions. Please see lines 112-132 in the updated manuscript.
>
>
> >Q. How robust is Hide and Seek to correlated features or spurious dependencies.
>
> A. Spurious dependencies do not cause a problem in case the data set is sufficiently large, as NN architectures such as Hide and Seek are surprisingly good at avoiding overfitting. Highly correlated features can become problematic as in general, establishing feature importance from observational data in the presence of highly correlated features is a challenging task. Also, if the relation between two features is (almost) deterministic, the hide-and-seek architecture may suffer from information leak. This can be avoided if the fake features are drawn conditioned on the important features. Please refer to the discussion in lines 300-320 in the updated manuscript.

---

### Official Review · Reviewer_Vbsn · 2025-10-31

**Soundness:** 1
**Presentation:** 2
**Contribution:** 2
**Rating:** 2
**Confidence:** 4

**Summary:**

This paper defines a notion of event-level causality, presents a method for learning it, and then uses it as a tool for explaining the outputs of AI systems. It compares this novel tool (that builds on an existing one) to existing causal feature importance tools on several use-cases, showing that it performs better.

**Strengths:**

The general idea of using causal models to improve local explanations of AI systems is a good one, and this paper aims to contribute to that idea.

The definition of event-level parenthood (if worked out properly) is potentially an interesting concept that deserves its place in the causal literature.

**Weaknesses:**

There are several major problems with the paper.

__A__: The formal details are not worked out well at all, and lead me to believe that they are either inconsistent, or far too convoluted.

__B__: The formal definitions are supposed to be motivated by the fact that we can use them for explaining an output Y of an AI method. However, the setting of section 4 is much more restricted than that of section 3, to such an extent that their novel definitions are entirely unnecessary.

__C__: Contrary to what the authors claim, their notion of event-level causality is not substantially different from actual causation.

__D__: Related to the above, there exists work that is extremely similar to the current approach, and yet it not discussed at all.

I will make these problems more concrete in the Questions section below.

__Some minor issues:__

__E__: There exists relevant prior work that could be mentioned. The idea of allowing for different causal structures for different realizations of variables is not novel. Firstly, when restricted to realizations of exogenous variables (or root variables), this idea appears in several works on the logic of causal models, see for example Halpern (2016) "Actual Causality", chapter 2. Secondly, the idea of having event-level parents was introduced already in Beckers (2021) "Equivalent causal models". (There the focus was exclusively on the logical setting, but the probabilistic setting can trivially be added on top of this.)

__F__: I find the example badly chosen. Time does not cause humidity, time does not cause anything at all. So it’s strange to take T as a causal variable. It would be much more natural to just use time-indexed variables for S and L.

**Questions:**

__A: Questions and comments regarding the formal details.__

__Issue 1:__

Although never made explicit, the authors assume that there is a fixed total ordering of the variables, i.e., an ordering that does _not_ depend on the realization. (Without such an ordering, their definitions are not consistent.) This should be made explicit.

It is not at all clear what the authors are trying to do with their definitions.

Here is the first interpretation of what they are trying to do. There is the standard, variable level causal Markov condition, with the accompanying joint distribution P, and the variable level parents. For any such distribution P, we can consider for any X, the variable level parents PaX, and any realization $PaX=pax$, the distribution $P(X | PaX= pax)$. In some cases, it holds that  $P(X | PaX= pax)=P(X | PaX^* = {pax*})$  for some $PaX^* \subset PaX$ and with $pax*$ the restriction of $pax$ to $PaX*$. In such a case, we can say that $PaX*$ form a set of event-level parents of X for the realization $PaX= pax$. By extension, it forms a set of event-level parents of X for any realization $X_<=x_<$ that extends $PaX= pax$ to a realization over all variables with lower index than X.

Note that there may exist multiple choices $PaX^*$ that are subset-minimal for which this is true. (For example, if $P(Y | A=1,B=1)=P(Y | A=1)=P(Y | B=1)$, then both $A$ and $B$ form a set of event-level parents of Y.) Because of this lack of unicity, the original DAG and distribution P does not suffice to define a function $H(x_{<i})$ that returns the event-level parents of $X_i$. And _that_ is why the authors introduce what is essentially an oracle function $H$ that results in a specific choice of event-level parents for each realization. The event-level DAGs are then nothing but the DAGs that are constructed along the ordering given by H.

The above interpretation is not in any way explicitly given in the paper. Rather, the paper presents something that seems much more general and much more complicated. This suggests that there is a second interpretation, which agrees with some parts of the first interpretation, but not all. I fail to see how this second interpretation works. We have a joint distribution P(X). Either this distribution allows for a variable-level Markov factorization, or it doesn't. Given that the authors assume that there is a fixed ordering of the variables, and given that all of the event-level Markov factorizations respect this ordering, we can use this ordering to give us a variable-level Markov factorization. But then it is wrong to present their assumption as being less restrictive, since it also relies on assuming a variable-level Markov factorization.

Rather, their framework is one which adds a further function H to the standard framework, that gives us a different Markov factorization for each realization. But this factorization has to respect the original, variable-level, factorization, doesn't it? So aren't we then left merely with H being allowed to select some subset of the original, variable-level parents, for which P tells us that these parents suffice? But then we are back at the first interpretation... Perhaps I am overlooking something, but then the authors should make this all much more clear, and give examples.

Let us assume that the first interpretation is correct. Then why not present their approach in the manner that I did? Surely that is much easier than the presentation given in the paper. Also, given the lack of unicity, it seems as if all of the hard work is being done by an oracle H. But then this overlooks the question as to what determines H, and that is the question we need to answer: what were the variables that explain some output variable? Perhaps the following issue holds the key to answering this?

__Issue 2:__

First H is explained, and then the authors switch to finding H*. I didn’t understand what to make of this. Is the idea that we are _constructing_ some H*? Or are we approximating some ground-truth H?

__B: formal definitions from Section 3 are not necessary.__

In section 4 the authors move to the application of their definitions. But the application doesn’t seem to require their complicated definitions: we fix the output variable Y, and it is the variable with highest index, so it appears that all of the information regarding event-level parents for all others variables is irrelevant. Rather, for a given realization X=x, we are simply looking for some small subset X*=x* that explains Y=y. The fixed ordering is entirely irrelevant, and the event-level DAGs are mostly entirely irrelevant as well: all we need, is to decide which members of X were actually relevant for Y. So the novelty of their approach disappears.

__C: Actual causality__

Relatedly, contrary to what the authors claim, in this restricted XAI setting their notion of event-level causality is the same as the notion of actual causality. The authors say it is different because actual causation is backwardlooking and their approach is not. This difference disappears entirely given that the only forwardlooking that occurs here is relative to Y, the final variable. In other words, the only candidate difference is that they focus on explaining the variable Y instead of a specific event Y=y. But even that small difference does not exist, for their method focusses on actual, specific, outcomes Y=y, for that is the only way they can compare to all the other methods, which do this. In other words, they compare to local XAI methods, and that which is to be explained is some Y=y, not the variable Y.

__D: closely related work__

They are not the first to invoke causal explanations for the purposes of local XAI, and they are not the first to compare these to other methods. In fact, given the above, their method is extremely similar to the one that is theoretically developed in Chockler and Halpern (2024) “Explaining Image Classifiers” and then implemented in several follow-up papers, the most relevant one being Chockler et al. (2025) “Causal explanations for image classifiers”. Furthermore, these papers prove that Halpern's definition of actual causality in this setting simplifies drastically, and seems to correspond pretty much to what the authors propose. (Similar results are to be found in Beckers (2022) "Causal explanations and XAI".)  So the authors should discuss this alternative approach, and explain what is novel about theirs.

__Some minor formal issues:__

193: Why speak of H* as a structure, when it is a function? And why not make the argument explicit? That is, we need to know the realization X=x before H* returns a set of parents, and yet that is not clear at all from the notation.

194: “the minimal set”: as mentioned, unicity is not guaranteed.

(6) What is the norm here?

(7) Why use different types of distributions for (a), (b), and (d)? Why not use the empirical distribution in all three cases?

---

> ### Author Response · Authors · 2025-11-22
> **Main response to Reviewer Vbsn**
>
> We thank the reviewer for their comments; however, it appears that there are several major misunderstandings.
>
> A: The reviewer states that our formal details are either not worked out or are too convoluted.
>
> In item (A) we show that this is not the case. Starting from the reviewer’s own notations and interpretation of our paper, we demonstrate that once their proposed notation is made slightly more formal and rigorous, it directly converges to the formulas presented in the paper.
>
>
> B: The reviewer states that because the setting of Section 4 is more restricted than that of Section 3, our novel definitions (in Section 3) are unnecessary.
> In item (B) we explain that the contribution of Sections 3 and 4 is precisely to show that, under appropriate assumptions, a nuanced form of structure learning reduces to a set of independent XAI problems (with particular forms). In other words, the reduction from the harder setting of event-level structure learning to the more restricted and simpler setting of XAI is itself a key contribution of our work.
> This reduction further motivates our design decisions in our second contribution, i.e., introducing a new deep learning architecture, “Hide and Seek,” designed specifically for that XAI setting. The reviewer has entirely overlooked this second contribution and does not mention it at all.
>
> C: The reviewer states that our notion of event-level causality is not substantially different from actual causation.
> In item (C), we discuss the differences.  In Appendix B of the updated manuscript, we also provide a detailed clarification of how the two approaches differ.
>
> D: The reviewer states that the work of Chockler et al. (2025) is “extremely similar” to ours, yet we do not discuss it.
>
> We thank the reviewer for bringing this work to our attention. In Appendix B of the updated manuscript, we now cite it (along with Beckers 2022) and discuss several key differences from our approach, highlighting that Chockler et al. (2025) is not closely related to our work. In contrast, our method shares much more in common with approaches such as INVASE and REAL-X.
>
> Chockler et al. (2025) can explain but cannot predict. It handles only discrete features. It applies only when the relationship between input features and the output is deterministic. It also requires collecting new data for each explanation instance, querying the black-box model for all possible feature values. Moreover, it relies on the assumption that all input features are causally independent. This assumption severely limits the applicability of that work and makes it immediately irrelevant to the problem of Bayesian Network structure learning, where the random variables are, by definition, allowed to be causally dependent. Finally, the cited work is purely logic-based and does not learn in any machine learning sense; it contains no optimization or deep learning components.
>
> In contrast, our proposed XAI framework has none of these limitations. Our approach learns to explain as well as to predict via a new deep-learning architecture. It can handle both discrete and continuous features. It applies whether the relationship between input and output is stochastic or deterministic. It learns purely from the available data and requires no interventions or active data collection. Crucially, our method does not assume causal independence among the input features, which is why, when the order of the random variables is known, it can be applied to event-level structure learning.
>
> Given these substantial differences, the claim that our work is “extremely similar” to that of Chockler et al. suggests that the reviewer has overlooked the main contributions of our paper.
>
> We hope that this rebuttal clarifies these points and enables a fairer assessment of our contributions and novelties.

---

> > ### Author Response · Authors · 2025-11-22
> > **A: On formal details -- Issue 1**
> >
> > **On the known order assumption**
> >
> > First, in response to the reviewer’s comment:
> >
> > >``Although never made explicit, the authors assume that there is a fixed total ordering of the variables, i.e., an ordering that does not depend on the realization.'',
> >
> > We note that this assumption is explicitly stated in lines 112–122 of the paper and is referred to as the “Known order assumption.”
> >
> > **On the derivation of the formulas**
> > Regarding the remaining concerns, we begin from the reviewer’s own “first interpretation” and show that, with only minor notational adjustments, it is equivalent to what is presented in the paper.
> >
> > We quote the Reviewer's ``first interpretation":
> >
> >
> > >``There is the standard, variable-level causal Markov condition, with the accompanying joint distribution $P$, and the variable-level parents. For any such distribution $P$, we can consider for any $X$, the variable-level parents $PaX$, and any realization $PaX=pax$, the distribution $P(X | PaX= pax)$. In some cases, it holds that $P(X | PaX= pax)=P(X | PaX^{\star} = {pax^{\star}})$ for some $PaX^{\star} \subset PaX$ and with $pax^{\star}$ the restriction of $pax$ to $PaX*$. In such a case, we can say that $PaX*$ forms a set of event-level parents of $X$ for the realization $PaX= pax$. By extension, it forms a set of event-level parents of $X$ for any realization $X_<=x_<$ that extends $PaX= pax$ to a realization over all variables with lower index than $X$.
> > "
> >
> >
> > The above interpretation is correct. It is expressed slightly differently, but as we will show, it is equivalent to our formalism.
> >
> >
> > The reviewer considers $PaX$ to be the "variable-level parents" of the random variable $X$.
> > For clarity, let's use the notation $X_i$ instead of $X$ and $PaX_i$ instead of $PaX$ where $i \in \\{1, ..., q\\}$ since $X := X_i \in \\{X_1, ..., X_q\\}$.
> > Moreover, instead of the reviewer's wording:
> >
> > >``In *some cases*, it holds that
> > $P(X | PaX= pax)=P(X | PaX^{\star} = {pax^{\star}})$ (Eq1)
> > >for some $PaX^{\star} \subset PaX$",
> >
> > let's be more precise and say:
> >
> > *it holds that*
> >
> > $P(X_i | PaX_i= pax_i)=P(X_i | PaX_i^{\star}(x_{<i}) = {pax_i^{\star}}(x_{<i}))$ (Eq2)
> >
> > That is, since for each realization $x_{<i}$ of $X_1$ to $X_{i-1}$, what the review calls $PaX^{\star}$ could potentially be a different subset, it is more precise to treat $PaX^{\star}$ as a function of $x_{<i}$, i.e. $PaX_i^{\star}(x_{<i})$.
> >
> > A subtle difference is that in our paper, we do not directly work with ``variable-level parents", $PaX_i$ (and we do not introduce a notation for it); instead, we use all preceding RVs, $X_{<i}$. This does not change any conclusions, and the reviewer's formula (Eq2) is equivalent to:
> >
> >
> > $P(X_i | X_{<i}= x_{<i})=P(X | PaX_i^{\star}(x_{<i}) = {pax_i^{\star}}(x_{<i}))$  (Eq3)
> >
> >
> > This equivalence holds
> > because $PaX_i \subseteq X_{<i}$ and any random variables in $X_{<i}$ that are not in $PaX_i$ are, by definition, statistically independent of $X_i$ given $PaX_i^{\star}$ and therefore do not affect the conditional distribution.
> >
> > Assuming that the reviewer accepts these minor notational adjustments in their first interpretation, let's denote the event-level parents of $X_i$ in the context $x_{<i}$ by $X_{pa_i}(x_{<i})$ (instead of $PaX_i^{\star}(x_{<i})$) and their realization by $x_{pa_i}(x_{<i})$ (instead of $pax_i^{\star}(x_{<i})$).
> > With this alternative notation,
> > the expression $PaX_i^{\star}(x_{<i}) = {pax_i^{\star}}(x_{<i})$ becomes
> > $X_{pa_i}(x_{<i}) = x_{pa_i}(x_{<i})$.
> > Note that here in the latter notation
> > ${pa_i}(x_{<i})$ is a function that takes a realization of the preceding random variables,$x_{<i}$, as input and returns the indices of the event-level parents of $X_i$.
> >
> >
> >
> > Given these notational changes, (Eq3) is restated as:
> >
> >
> > $P(X_i | X_{<i})= P(X_i | X_{pa_i (x_{<i})} = x_{pa_i (x_{<i})}  )$  (Eq4).
> >
> > By multiplying all the factors, we obtain the joint distribution:
> >
> >
> > $P(X_{1:q} = x_{1:q}) := \prod_{i=1}^q P(X_i=x_i | X_{<i})= \prod_{i=1}^q P(X_i |
> > X_{pa_i (x_{<i})} = x_{pa_i (x_{<i})}
> > )$ (Eq5)
> >
> >
> > which is equivalent to equation (4) in the main text (except that in (4) we also have a superscript $(\mathcal{H})$ to highlight the lack of uniqueness -- For further discussion, see Issue 2)
> >
> > In short, our formal details are neither "inconsistent" nor "too convoluted" nor "more general and much more complicated". They are explicitly provided in the paper and are equivalent to the reviewer's first interpretation of the text, with only minor notational adjustments to make the representation more precise and rigorous.
> >
> >
> > Given this clarification, we hope the reviewer recognizes that their criticism regarding the paper's formal details is not justified.

---

> > > ### Author Response · Authors · 2025-11-22
> > > **A: On formal details -- Issue 2**
> > >
> > > **On $\mathcal{H}$ and $\mathcal{H}^{\star}$**
> > >
> > > The factorization of a joint distribution is not necessarily unique. For each node, there might be multiple parenthood functions that satisfy the event-specific conditional independence. To differentiate them, we use superscripts such as $\mathcal{H}$ (see line 160-161).
> > > We call $\mathcal{H} := (pa_1^{\mathcal{H}}, ..., pa_q^{\mathcal{H}})$ a "structure" since in the classical structure learning a graph $\mathcal{G} := (pa_1^{\mathcal{G}}, ..., pa_q^{\mathcal{G}})$ represents the structure of the Bayesian Network (see line 87).
> > > The difference is that in classic structure learning, $pa_i^{\mathcal{G}}$ is a fixed set of indices corresponding to the parents of node $X_i$, whereas in our setting $pa_i^{\mathcal{H}}$ is a function that takes a context $x_{<i}$ and returns a set of parent indices specific to that context. Therefore, while $\mathcal{G}$ is a single directed acyclic graph (DAG), $\mathcal{H}$ induces a potentially different DAG for each event $x_{1:q}$.
> > >
> > > For example, in the Reviewer’s proposed scenario (where $A$, $B$ and $Y$ are renamed to $X_1$, $X_2$ and $X_3$),
> > > let $\mathcal{H}$, $\mathcal{H}'$ and $\mathcal{H}''$ be three event-level causality structures and  $pa_{3}^{(\mathcal{H})}(X_1=1, X_2=1) = \{1\}$ (indicating that for the realization/context $(X_1=1, X_2=1)$, according to $\mathcal{H}$, the parent of $X_3$ is $X_1$), $pa_{3}^{\mathcal{H}'}(X_1=1, X_2=1) = \{2\}$ (i.e. the parent is $X_2$) and
> > > $pa_{3}^{\mathcal{H}''}(X_1=1, X_2=1) = \{1, 2\}$ (i.e. parents are $X_1$ and $X_2$).
> > > What we look for is the most parsimonious such structure $\mathcal{H}^{\star}:= (pa_1^{\mathcal{H}^{\star}}, pa_2^{\mathcal{H}^{\star}}, pa_3^{\mathcal{H}^{\star}})$ (See equation (6) in the main text). In the above example, $\mathcal{H}^{\star}$ is not unique, and our proposed deep learning mechanism can either learn that, $pa_3^{\mathcal{H}^{\star}} = pa_3^{\mathcal{H}}$, or $pa_3^{\mathcal{H}^{\star}} = pa_3^{\mathcal{H'}}$.

---

> > > > ### Author Response · Authors · 2025-11-22
> > > > **B. On the necessity of the formal definitions in Section 3**
> > > >
> > > > The reviewer correctly notes that in Section 4, where we focus on the XAI task, the problem is simpler than in Section 3 and that a fixed ordering is not required for the XAI task.
> > > >
> > > > However, the conclusion that
> > > > >“the novelty of our approach disappears’’
> > > >
> > > > does not follow from these observations.
> > > >
> > > > We have never claimed that the "known order assumption" is required for XAI.
> > > > Instead, the central message of Sections 3 and 4 is that, under the known-order assumption, the Bayesian network structure learning problem reduces to $q-1$ independent XAI problems of the form Eq. (8) in the main text, provided that the structure captures event-level rather than RV-level dependencies.
> > > >
> > > > This reduction, from structure learning to XAI, is a key contribution of the paper and establishes a conceptual bridge between the two fields. The reviewer appears to have overlooked this point.
> > > > To clarify the logical flow of the paper, we summarize the contributions below:
> > > >
> > > > (I) While different XAI methods define “feature importance’’ in different ways, Sections 3 and 4 motivate using the definition stated by Eq. (8) because it is the only feature-importance definition that unifies structure learning and XAI.
> > > >
> > > > (II) Section 5 then introduces a new deep learning architecture (Hide and Seek) explicitly designed to learn Eq. (8) effectively from observational data.
> > > > It is worth noting that some existing XAI approaches, such as INVASE and REAL-X, also define feature importance via Eq. (8) and attempt to learn it from observational data.
> > > >
> > > > However:
> > > >
> > > > (a) they do not provide the conceptual motivation that we established in Sections 3 and 4, and
> > > >
> > > > (b) They suffer from limitations such as information leakage and do not learn the objective function as effectively as our Hide and Seek architecture (see Section 4.1).

---

> > > > > ### Author Response · Authors · 2025-11-22
> > > > > **C. On the relation with Actual Causality**
> > > > >
> > > > > The reviewer states that when focusing on the parenthood function of a single node (i.e., the XAI setting), the difference between our approach and actual causality disappears.
> > > > >
> > > > > We disagree: even in this restricted setting, event-level causality and actual causality remain different concepts.
> > > > >
> > > > > To start with, if the actual causality is defined as in Chockler et al. (2025), then the relation between the inputs and the output must be deterministic; otherwise, expressions such as $(M, u) \models [X \leftarrow x] \neg \varphi$, are not valid logical statements.
> > > > > This immediately separates their setting from ours: event-level causality is inherently stochastic.
> > > > >
> > > > > Even if we extend the definitions of actual causality to a probabilistic form, one must still assume causal independence of inputs; without this assumption, variable assignments, $X=x$, and interventions, $[X \leftarrow x]$, are not equivalent. This causal-independence requirement is highly restrictive and greatly limits the applicability of actual causality.

---

> > > > > > ### Author Response · Authors · 2025-11-22
> > > > > > **Minor Issues**
> > > > > >
> > > > > > Q. "193: Why speak of $\mathcal{H}^{\star}$ as a structure, when it is a function? And why not make the argument explicit? That is, we need to know the realization $X=x$ before $\mathcal{H}^{\star}$ returns a set of parents, and yet that is not clear at all from the notation."
> > > > > >
> > > > > > A. $\mathcal{H}^{\star} := (pa_1^{\star}, ..., pa_q^*)$ (defined in line 195-6) is not a single function. It is a tuple of the parenthood functions, one for each node. Each individual parenthood function is a function and we have made this explicit by notation $pa_i(\textbf{x}_{<i})$.
> > > > > >
> > > > > > Q. "(6) What is the norm in($||pa_i(\textbf{x}_{<i}) ||$)?"
> > > > > >
> > > > > > A. The number of parents of the $X_i$ when the preceding variables take the value $\textbf{x}_{<i}$.
> > > > > >
> > > > > > Q. "(7) Why use different types of distributions for (a), (b), and (d)? Why not use the empirical distribution in all three cases?"
> > > > > >
> > > > > > A. An empirical distribution is an approximation of a true distribution, obtained by a finite set of samples.
> > > > > > These are conceptually different objects and therefore require different notation.
> > > > > > For example, the expected value of a function $f$ w.r.t. a true distribution $p(X)$ , that is $\mathbb{E}_{p(X)}$ is equal to
> > > > > > $\int p(x) f(x) dx$.
> > > > > >
> > > > > >
> > > > > > If $\widetilde{p}(X) = \\{ x^{(1)}, ... , x^{(N)} \\}$ is the empirical distribution consisting of $N$ independent draws from $p(X)$, then the expected value of $f$ w.r.t. $\widetilde{p}(X)$ is
> > > > > > $\sum_{t=1}^N f(x^{[t]})$.

---

### Official Review · Reviewer_6A4B · 2025-10-31

**Soundness:** 2
**Presentation:** 2
**Contribution:** 3
**Rating:** 4
**Confidence:** 3

**Summary:**

Context-specific (here called "event-level") independence is used to define "event-level causality", which is then used to define a notion of causal feature importance. A two-part neural network architecture is proposed to select the causally important features.

**Strengths:**

- Context-specific independence can be a valuable tool in explainable AI, and this paper contributes a generally applicable, end-to-end differentiable approach "Hide&Seek" to learn such independences from data.

**Weaknesses:**

- Early parts of the paper discuss Bayesian network structure learning and causal discovery. However, in the context of the paper, there is a known topological ordering of the variables, in which case the aforementioned problems reduce to feature selection (and indeed, this reduction is made at the start of section 4). I think the message would be clearer if feature selection was discussed from the start. I am also not convinced that this paper manages to "unify" causal modelling and explainable AI, as claimed in the early sections.

**Questions:**

- In section 4 (and the appendix), it is shown that an earlier approach may suffer from an information leak. But I didn't see a proof in section 5 that the newly introduced method does not suffer from such a leak. Do you believe this to be true?
- "the minimal set of parents" (line 194) may not be unique! For instance if $X_1$ and $X_2$ are independent coin flips and $X_3 = X_1 \wedge X_2$. Which feature should then be called causally important in the different realizations? And does the Hide&Seek algorithm handle this correctly?
- line 212-213: choosing $\lambda$ this way does *not* ensure what is claimed, because a too small $\lambda$ will lead to overfitting. How would you address this?

### Other remarks
- "Bayesian structure learning" has a different meaning than "Bayesian network structure learning", but they are used interchangeably in this paper, and both abbreviated to BSL. I suggest being consistent and always using BNSL.
- line 79: final index i should be k
- line 188-189: "that precede $X_i$ (for the realization $x_{<i}$)" suggests that the predecessor relation may change depending on the realization, but that's not what this is trying to say
- equation (6): the left-hand side should be $pa_i^*(x_{<1}$, because the right-hand side also depends on $x_{<1}$
- equation (6): set size is usually denoted by single, not double vertical bars

---

> ### Author Response · Authors · 2025-11-25
> **Response to Reviewer 6A4B (on Weaknesses and Q1)**
>
> We thank the reviewer for their time and effort in reviewing our paper.
>
> **Regarding the Weaknesses:**
> Showing that event-level structure learning reduces to a set of independent XAI problems is one of the contributions of our work.
> Sections 2 and 3 motivate our choice to approximate Equation (8) as the XAI objective, rather than adopting alternative definitions of “feature importance’’ (e.g., those used in LIME or Gradient-based XAI methods).
>
>
> >Q. It is shown that an earlier approach may suffer from an information leak. But I didn't see a proof in section 5 that the newly introduced method does not suffer from such a leak. Do you believe this to be true?
>
> A. Thanks for asking this question. This is a core contribution of our work, and we should explain it clearly.
>
> The source of the information leak in ablation-based existing methods is that, through the ablation mask, 1 bit of information is transmitted from the "selector" to the  "predictor" per ablated feature.
>
> If there is no way for the "predictor" (in our work, the "Seek" module) to differentiate between a signal, $x \sim P(X)$, drawn directly from the original joint distribution and a modified signal, $z=\mathcal{F}(x, x^{fake})$, coming through the "selector" (i.e. our "Hide" module), then no information leak is possible.
>
> In other words, the modified signal, $z$, should also be distributed according to $P(X)$.
>
> (Note: $x$ and $z$ are vectors, but unlike in the manuscript, we do not display the vectors in boldface here to avoid markdown rendering issues.)
>
> Let $S$, short for $S(x)$, be the set of indices of RVs that are selected as important features by the "selector" (and therefore passed through); and let $\bar{S}$ denote the indices of the non-selected RVs (which are replaced by fake values).
> By lines 274-277 of the text:
>
> $z = (z_S, z_{\bar{S}}) = (x_S, x_{\bar{S}}^{fake})$.
>
> To ensure $P(z) = P(x)$, we need to draw  $x_{\bar{S}}^{fake}$ from $P(x_{\bar{S}} | x_S)$.
>
> The reason is that
>
> $P(z) =  P(x_S, x_{\bar{S}}^{fake}) = P(x_S)P(x_{\bar{S}}^{fake} | x_S)$
>
> is equal to $P(x) = P(x_S)P(x_{\bar{S}} | x_S)$,
> if and only if
>
> $P(x_{\bar{S}}^{fake} | x_S) =
> P(x_{\bar{S}} | x_S)$.
>
> This is the theoretical condition that guarantees no information leak occurs.
>
> For simplicity, in our work, $x_{\bar{S}}^{fake}$ is drawn from  $P(x_{\bar{S}})$.
>
> So **if** the "predictor" can differentiate between a draw from
> $P(x_{\bar{S}} | x_S)$ and a draw from
> $P(x_{\bar{S}})$, this could be a potential source of information leak.
>
> Nonetheless, this is extremely unlikely in practice.
> The only exception is when the marginal $P(x_{\bar{S}})$ and the conditional $P(x_{\bar{S}} | x_S)$ are significantly different i.e. a subset of
> $x_{\bar{S}}$ and a subset of $x_S$ are strongly correlated.
>
> However, this particular setting represents an extremely challenging scenario for any XAI approach, as it would be very hard to correctly determine which of the highly correlated features is the true cause or explanation.
>
> We had originally included these points in a footnote; however, in response to the reviewers’ feedback, we have now moved the formal proof (of the sufficient condition that prevents information leak) as well as an expanded discussion on how we approximate it into the main text (see lines 300–320 of the updated manuscript).

---

> ### Author Response · Authors · 2025-11-25
> **Response to Reviewer 6A4B (Q2 and Q3)**
>
> >Q. "The minimal set of parents" may not be unique! For instance if $X_1$ and $X_2$ are  are independent coin flips and $X_3 = X_1 \wedge X_2$, which feature should then be called causally important in the different realizations? And does the Hide and Seek algorithm handle this correctly?
>
> A. That's true. The minimal set of parents is not necessarily unique. In the above example,
> for the events $(X_1=1, X_2=1)$,  $(X_1=1, X_2=0)$ and $(X_1=0, X_2=1)$, the correspoinding minimal set of event-level parents is unique and equal to $\\{X_1, X_2\\}$, $\\{X_2\\}$ and $\\{X_1\\}$, respectively.
>
> However, as the reviewer pointed out, for the event $(X_1=0, X_2=0)$, the minimal set of event-level parents is *not* unique as either $\\{X_1\\}$ or $\\{X_1\\}$ satisfies (Eq. 8), as conditioning on either $(X_1=0)$ or $(X_2=0)$ makes $X_3$ independent of the remaining RV.
>
> If the minimal set of event-level parents is unique, Hide and Seek correctly identifies it.
> For the event, $(X_1=0, X_2=0)$, Hide and Seek will return either $\\{X_1\\}$ or $\\{X_2\\}$, but not their union, which is consistent with our definition. In this case, the loss function (Eq. 11) of the Hide and Seek network has two equivalent minima, and depending on initialization, gradient descent will converge to one of them.
>
> We have clarified the lack of uniqueness in lines 212-215 of the updated manuscript.
>
>
> >Q. line 212-213: choosing $\\lambda$ this way does not ensure what is claimed, because a too small $\\lambda$ will lead to overfitting. How would you address this?
>
> A. Thank you for raising this point. In those lines, the assumption is that (a) removing any subset of the true parents increases the divergence by at least $\\epsilon$ and (b) removing any RV that is not a true parent has no effect on the divergence. This setting is intended just to motivate that, under the appropriate assumptions, the loss functions (6) and (7) have the same minimizer; As such, it is reasonable in practice to approximate (6) by (7).
> From this motivation, we cannot conclude that $\\lambda$ can be arbitrarily small.
> The reason is that in practice, assumption (b) does not hold, and due to spurious correlations and overfitting, the NN-based model can potentially learn to predict the output from non-parent RVs.
>
> We have clarified this in lines 232-236 of the updated manuscript.
>
> In practice, $\\lambda$ is just a hyperparameter, and we have fixed it for all our experiments. (lines 353-355 of the updated manuscript).
>
>
> **Final remarks.**
> We thank the reviewer again for their thoughtful comments. They highlighted parts of the paper that needed more explanation and expansion.
>
> We hope that our clarifications of the raised points and reflecting them in the updated manuscript have satisfactorily addressed their concerns.

---

### Meta-Review · Area_Chair_L2Pq · 2026-01-03

**Summary:**

Reviewers raised concerns that the proposed notion of “event-level causality” is formulated via observational factorization and conditional independence, but lacks a clear connection to intervention-based or mechanism-based causal semantics. As a result, the learned parenthood functions are more naturally interpreted as instance-wise predictive feature selection rather than as causal parents. Reviewers also questioned the necessity and role of the formal framework in Section 3, noting that the reduction to XAI does not support the claimed causal interpretation. In addition, several formal claims about identifiability and required assumptions (e.g., known orderings or interventions) were considered overstated or insufficiently justified.

**Reviewer Concerns:**

Main concerns

1. Formulation and causal status of “event-level causality”

The proposed notion of event-level causality is formulated via realization-dependent conditional independence and an associated factorization of the observational joint distribution. Constructing such a factorization is mathematically valid, but by itself it does not establish causal semantics in the standard sense (i.e., relating to interventions, causal mechanism). In particular, the learned “parenthood functions” are characterized through predictive sufficiency of conditioning sets, and therefore are most naturally interpreted as instance-wise minimal sufficient predictors under the observational distribution, rather than as causal parents.

The above highlights the distinction between Bayesian network and causal modeling.

This core point is not addressed in the rebuttal.

2. Role and necessity of Section 3 relative to the claimed contributions

Given the above, Reviewer Vbsn’s concern about the necessity of Section 3 is well-founded. The reduction argued by the authors， mapping event-level structure learning (under a known ordering) to a collection of instance-wise feature selection objectives, may be a useful conceptual connection between Bayesian factorization and XAI. However, this reduction alone does not provide causal interpretation: it remains within the realm of observational factorization and predictive conditioning. Consequently, the algorithmic contribution is best viewed as an instance-wise feature selection / explanation method (learned from observational data), rather than a causality-based framework.

3. Formal clarity and assumptions

Several formal aspects remain problematic or at least insufficiently justified:

a. The statement that Causal Sufficiency + Causal Markov + Faithfulness allow recovery of the true causal graph is, in general, too strong; under these assumptions , one typically recovers a Markov equivalence class (e.g., a CPDAG), unless additional assumptions (functional form, non-Gaussianity, interventions, heterogeneity across environments, etc.) are imposed.

b. "To identify the true causal graph, either there **must** be a known partial topological ordering or sufficient interventional data should be available", Note: neither known ordering nor interventional data is necessary for identifiability, they are sufficient under certain conditions, but not necessary in general. There exist well-established observational methods that recover causal directions under additional assumptions (e.g., functional form, heterogeneity across environments), without relying on prior orderings or interventions.

.....

Overall, the paper contains interesting ideas and may provide a useful XAI-style instance-wise feature selection approach. However, the central causal interpretation is not sufficiently supported by the current formalization and assumptions. For these reasons, I recommend rejection.

**Reviewer Scores:**

The core conceptual and formal concerns raised by Reviewer Vbsn (in particular regarding the causal interpretation of the framework, the necessity of the formalism in Section 3, and the relation to actual causation) were not substantively resolved in the rebuttal. While some clarifications and additional references were added, the central issues remain. I therefore believe this reviewer would likely maintain their original score.

These issues also underlie concerns raised by other reviewers. Reviewer 6A4B noted that the “minimal set of parents” (line 194) may not be unique, raising questions about which features should be considered causally important. Reviewer azxR raised related concerns about identifiability.

Given the fundamental issues, I do not expect the reviewers’ assessments to change substantially after discussion.

Although Reviewers rgxT and r2c6 gave more positive scores, these fundamental conceptual and formal concerns remain central to the paper’s main claims.

---

### Decision · Program_Chairs · 2026-01-26

Reject